# POEMetric: The Last Stanza of Humanity

**Bingru Li**
Department of Linguistics and Communication
University of Birmingham
`bxl426@student.bham.ac.uk`

**Han Wang**[*]
Department of Information Engineering
and Computer Science
University of Trento
`han.wang@unitn.it`

**Hazel Wilkinson**
Department of English Literature
Institute for Data and AI (IDAI)
University of Birmingham
`h.j.wilkinson@bham.ac.uk`

## Abstract

Large Language Models (LLMs) can compose poetry, but how far are they from human poets? In this paper, we introduce POEMetric, the first comprehensive framework for poetry evaluation, examining 1) basic instruction-following abilities in generating poems according to a certain form and theme, 2) advanced abilities of showing creativity, lexical diversity, and idiosyncrasy, evoking emotional resonance, and using imagery and literary devices, and 3) general appraisal of the overall poem quality and estimation of authorship. We curated a human poem dataset - 203 English poems of 7 fixed forms annotated with meter, rhyme patterns and themes - and experimented with 30 LLMs for poetry generation based on the same forms and themes of the human data, totaling 6,090 LLM poems. Based on POEMetric, we assessed the performance of both human poets and LLMs through rule-based evaluation and LLM-as-a-judge, whose results were validated by human experts. Results show that, though the top model achieved high form accuracy (4.26 out of 5.00, with Gemini-2.5-Pro as a judge; same below) and theme alignment (4.99), all models failed to reach the same level of advanced abilities as human poets, who achieved unparalleled creativity (4.02), idiosyncrasy (3.95), emotional resonance (4.06), and skillful use of imagery (4.49) and literary devices (4.67). Humans also defeated the best-performing LLM in overall poem quality (4.22 vs. 3.20). As such, poetry generation remains a formidable challenge for LLMs. Data and codes are released at https://github.com/Bingru-Li/POEMetric.

## 1 Introduction

Large Language Models (LLMs) Hurst et al. (2024); Grattafiori et al. (2024); Team (2025b); Anthropic (2024); Guo et al. (2025); Qwen et al. (2025) have demonstrated outstanding capabilities in reasoning and logic tasks, ranging from solving math problems to coding. Nevertheless, less attention has been allocated to LLMs' abilities in terms of arts and humanities, let alone advanced tasks such as literary writing. Among the various literary forms, poetry, has long stood as the ultimate testament to linguistic artistry, demanding the perfect synthesis of verbal precision, emotional resonance, and cultural literacy within constrained forms. As such, compared with other forms such as essays and fictions, the compact and formulaic style of poetry makes it a valuable lens through which we are able to examine the generative abilities of LLMs.

While LLMs have excelled in text generation across numerous domains, the generation of authentic poetry remains a challenge. Though extant literature (e.g., Belouadi & Eger (2023); Ling & Zhang (2022); Yu et al. (2024)) has demonstrated the high formal accuracy in the meter and rhyme patterns of LLM-generated poems, there is still a lack of creativity and diversity Walsh et al. (2024b); Chen

---

[*]Corresponding author.

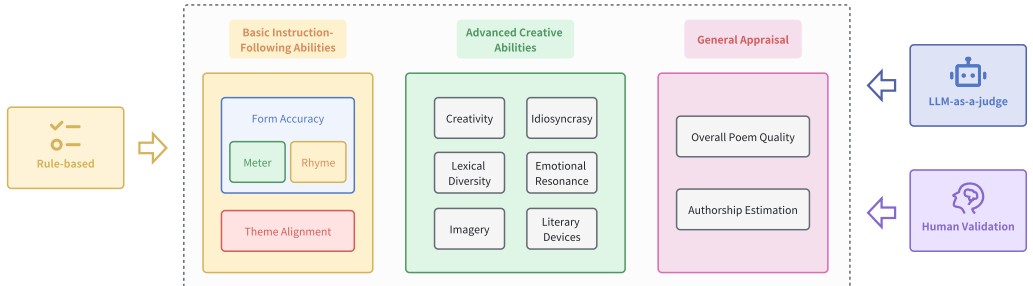

Figure 1: POEMetric. It comprises 10 metrics, including 1) basic instruction-following abilities (form accuracy and theme alignment), 2) advanced creative abilities (creativity, lexical diversity, idiosyncrasy, emotional resonance, and use of literary devices and imagery), and 3) general appraisal (overall poem quality and authorship estimation). Human- and LLM-authored poems are compared through rule-based evaluation and LLM-as-a-judge, whose results are validated by human experts.

et al. (2024). Moreover, little has been explored in terms of evaluating the artistic beauty as well as author intentions and emotions in the poems generated by LLMs, which are in fact the essence of poetry writing Greene et al. (2012). Therefore, the central inquiry of this paper lies in whether state-of-the-art LLMs can transcend mere syntactic competence to achieve what T.S. Eliot termed "the auditory imagination" Eliot (1986) - the fusion of sound, sense, and cultural memory that distinguishes enduring poetry from mere grammatical arrangement.

To address these issues, we propose POEMetric, the first comprehensive metrics for the evaluation of poetry, which examines 1) basic instruction-following abilities (form accuracy and theme alignment), 2) advanced creative abilities (creativity, lexical diversity, idiosyncrasy, emotional resonance, and use of literary devices and imagery), and 3) general appraisal (overall poem quality and authorship estimation). To the best of our knowledge, POEMetric is so far the most comprehensive evaluation framework for poetry, making up for what has been lacking in previous metrics in terms of poetic beauty, personal characteristics, and emotional effects. Based on POEMetric, we compared human-written and LLM-generated poems through rule-based evaluation, with a self-written algorithm for automated form detection, and LLM-as-a-judge (Gemini-2.5-Pro), whose results were validated by human experts. An illustration of POEMetric is shown in Figure 1.

We also curated a human poem dataset, comprising 203 human English poems of 7 fixed forms, which spans the past 200 years and ranges from canonical works to less-known recent creations. According to the same form and themes in the human data, we prompted 30 LLMs for poetry generation. Evaluation results showed that, though top LLMs were able to achieve high scores in terms of form accuracy and theme alignment - for example, Gemini-2.5-Pro topped at 4.26 and 4.99 out of 5.00 (with Gemini-2.5-Pro as a judge; same below) - they still struggled to attain the same level of advanced creative abilities as human poets, where the latter excelled in creativity (4.02), idiosyncrasy (3.95), emotional resonance (4.06), and skillful use of imagery (4.49) and literary devices (4.67). Human poets also defeated the best-performing LLM, i.e., DeepSeek-R1, in terms of overall poem quality, at 4.22 vs. 3.20. While both evaluators could recognize some original poems, human poems remained markedly distinguishable from LLM verse, with distinct patterns emerging in areas such as emotional resonance and idiosyncratic use of language. The agreement among rule-based evaluation, LLM-as-a-judge and human experts validates the effectiveness of POEMetric.

To sum up, our contributions can be summarized as three-fold:

- we propose POEMetric, the first comprehensive framework for poetry evaluation, covering basic instruction-following abilities, advanced creative abilities, and general appraisal;

- we curated a human poem dataset, carefully annotated with the forms (including meter and rhyme patterns), themes, and imagery;

- we designed an algorithm to automatically detect the formal patterns of poems. We have provided the code and the public-domain human poem dataset as supplementary materials to ensure reproducibility.

## 2 RELATED WORKS

**Poetry generation with Language Models**   Some attempts have been made to train Language Models (LMs) to generate poetry that adheres to formal constraints such as patterns of meter, rhyme, and style. For example, ByGPT5 (Belouadi & Eger, 2023), PoeLM (Ormazabal et al., 2022), GPoet (Popescu-Belis et al., 2023), and a GPT-2-based model (Possi et al., 2024) integrated structural metrics such as rhyme and meter into generation. (Bena & Kalita, 2020) fine-tuned GPT-2 to express and elicit emotions in poems.

Language-specific adaptations have yielded high-quality poetry in low-resource languages (e.g., Pashto(Ullah et al., 2024), Arabic (Alyafeai et al., 2023; Beheitt & HajHmida, 2023), Vietnamese (Huynh & Bao, 2024), Czech (Chudoba & Rosa, 2024) and culturally nuanced styles (e.g., classical Chinese poetry (Ling & Zhang, 2022; Yu et al., 2024; Wang et al., 2016; Yi et al., 2017; Zhang et al., 2017; Liu et al., 2018; Yi et al., 2018; Liao et al., 2019; Liu et al., 2019; Yang et al., 2023; Fang, 2024), limericks (Lo et al., 2022), and Homeric poetry (Lamar & Chambers, 2019)). However, models struggle with stylistic variation and creativity (Walsh et al., 2024b; Chen et al., 2024; Cao & Cheng, 2024).

**Evaluation of poetic quality**   Combining objective metrics (e.g., meter and rhyme accuracy, BLEU (Beheitt & HajHmida, 2023; Liu et al., 2019), perplexity (Ormazabal et al., 2022; Liu et al., 2019)) and human judgments have provided robust evaluation. More recent metrics include ProFTAP (Deng et al., 2024) which adopted Turing-test-inspired frameworks to evaluate poetic indistinguishability from human works. Others (Yu et al., 2024) applied LLM-as-a-judge in evaluating LLM-generated poems, examining fluency, meaning, coherence, relevance, and aesthetics. Still others fine-tuned LMs for evaluation, as in (Sawicki et al., 2023) who fine-tuned GPT-3 to classify if an LLM-generated poem was written in the style of Whitman. In addition, diversity evaluations revealed gaps in semantic and formal variance and artistic creativity compared to human-written poetry (Walsh et al., 2024b; Chen et al., 2024).

To sum up, extant poem evaluation metrics are limited to meter and rhyme accuracy and formal diversity, or overly general aspects of text generation such as fluency and coherence, whereas more advanced and nuanced abilities are at the essence of poetry composition. As LLMs have proven competitive in writing in certain poetic forms, metrics that look at more advanced, poem-specific abilities in such areas as creativity, author intentions and emotions, and poetic aesthetics such as use of imagery and literary devices (Greene et al., 2012) which are particular poetic features, are urgently required; this is where our POEMetric comes into play.

## 3 THE HUMAN POEM DATASET

In this section, we report on how we collected the human poem dataset, covering 7 poetry forms. An elaboration on the features of these forms can be found in Appendix A. Our focus on fixed-form poetry was designed to address a fundamental challenge in creative evaluation: ensuring the benchmark is rigorous and diagnostic. By first evaluating poetry within these constrained forms, we establish a quantifiable baseline that is crucial for systematically developing and validating the more subjective metrics needed for the ambiguous challenge of free-verse poetry.

Following previous research (Walsh et al., 2024b;a), we collected the poems from two online databases, the Poetry Foundation[1] and the Academy of American Poets[2], totaling 1,309 poems. Due to the fact that not all human poems were strictly written according to a typical meter and rhyme pattern, we designed an algorithm to detect the meter and rhyme patterns for each poem, and only kept those that followed a certain prosodic pattern. In the end, the human dataset comprises 203 poems, which includes 95 ballads, 9 ghazals, 6 limericks, 3 pantoums, 7 sestinas, 71 sonnets, and 12 villanelles, as shown in Table 1. The time frame of this dataset spans from as early as the 1800s to the present time, including both well-known artworks written by famous poets as well as little-known or newly released poems. We also annotated the themes of the poems by drawing on public analysis (e.g., Poem Analysis (web, a) and Poem Hunter (web, b)), and the imagery used by designing a list of common imagery in English poems. An example of the human poem data is illustrated in Figure 2.

---

[1] https://www.poetryfoundation.org
[2] https://poets.org

Table 1: The distribution of human poems by form and source

| Source × Form | Ballad | Ghazal | Limerick | Pantoum | Sestina | Sonnet | Villanelle |
|---|---|---|---|---|---|---|---|
| Poetry Foundation | 86 | 4 | 6 | / | 6 | 61 | 10 |
| Academy of American Poets | 9 | 5 | / | 3 | 1 | 10 | 2 |

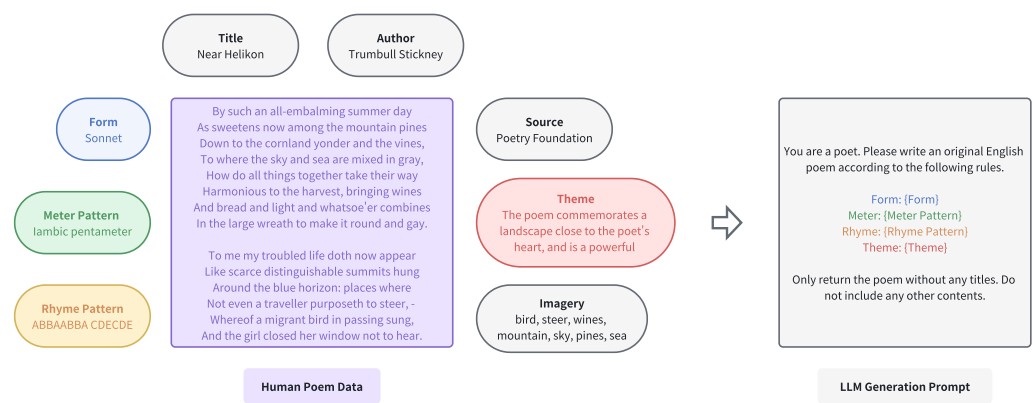

Figure 2: An example of the human poem data and the generation prompt for LLMs. On the left are the related data annotated about a poem, including author, title, poem content, source, form, meter pattern, rhyme pattern, theme, and imagery. Based on these data, the prompt for LLMs to generate poems is curated in the template on the right.

## 4 POEMETRIC

The paradigm of POEMetric is shown in Figure 1. The main part of the figure illustrates the 10 dimensions proposed, ranging from basic instruction-following abilities to advanced creative abilities and general appraisal, which will be discussed in 4.1. These dimensions are deeply rooted in literary theories and have been important in literary critique of poems (Greene et al., 2012) - the 6 dimensions in advanced creative abilities in particular - and yet overlooked by previous studies, as reviewed in Section 2. We also apply both objective and subjective evaluation techniques to triangulate our methodology, including a handcrafted algorithm as a quantitative metric, and LLM-as-a-judge and human experts for the more nuanced evaluation. Details are presented in Section 4.2.

### 4.1 THE DIMENSIONS OF POEM EVALUATION

**Basic instruction-following abilities** These include the examination of how well a poem is written in response to the given prompt, specifically in terms of the extent to which they follow the instructions on the **form**, including **meter** and **rhyme** where applicable, and the **theme** of a poem.

**Advanced creative abilities** POEMetric systematically and quantitatively applies the core, often qualitative, elements from traditional literary criticism to the more sophisticated evaluation of poetry generated by LLMs. To evaluate the more advanced abilities in poem creation, we assess the creativity, lexical diversity, idiosyncrasy, emotional resonance, and the use of literary devices and imagery. **Creativity** looks at whether the poem is written in a novel and creative way. **Lexical diversity** measures if the poem uses a varied vocabulary. Whether a poem demonstrates personal characteristics of the author is measured by **idiosyncrasy**, and whether a poem evokes **emotional resonance** is also examined. **Literary devices** are commonly used in human poems, and here we evaluate four typical techniques, i.e., simile, metaphor, personification, and allusion. The use of **imagery** shows if a poem can trigger a vivid image and engage the readers' senses. These 6 dimensions for advanced creative abilities that we have chosen are intended as a distillation of the features on which literary critics typically focus in the analysis of poetry, often known as 'Practical Criticism' (Richards, 2014).

**General appraisal**  Apart from the above fine-grained metrics, we also ask two more general questions. The first is to ask if a poem is good or not to evaluate its **overall quality**, and the second is to estimate the **authorship** of the poem, i.e., by a human or an LLM, which aims to explore to what extent the evaluators can distinguish between human-written and LLM-generated poems.

## 4.2 THE METHODS OF POEM EVALUATION

To make the framework more robust, we triangulate LLM-as-a-judge with rule-based quantitative evaluation and human expert judgments, as detailed below.

**Rule-based automated evaluation**  We applied a handcrafted, rule-based algorithm to automatically detect the meter and rhyme patterns in each poem in order to gauge the overall accuracy of each author. A flowchart of the algorithm can be found in Appendix B. For both human and LLM poems, lexical diversity is calculated with Moving Average Type-Token Ratio (MATTR) averaged across poems for each author, and creativity is quantified as the ratio of repetition of words in an LLM poem compared to the original human work, which is also averaged across poems for each author.

**LLM-as-a-judge automated evaluation and human validation**  To balance the need for large-scale evaluation with the practical constraints of high-quality literary analysis, we did not perform human validation on the entire dataset, which would otherwise be resource-intensive. The required evaluators - domain experts such as poets and literary academics - are a scarce resource. Furthermore, the annotation of a single poem is a highly demanding and time-consuming task, far exceeding the complexity of standard data-labeling. Therefore, our methodology leverages LLM-as-a-judge for broad coverage, complemented by the validation from a panel of human experts on a smaller, representative sample to ensure the reliability of the automated results.

We first provided all the anonymized LLM poems and human poems for LLM-as-a-judge for evaluation based on the dimensions discussed in 4.1. In order to validate the results, with Institutional Review Board (IRB) approval, we recruited 7 expert human judges to evaluate a subset of anonymized poems (58 in total) by humans and 7 representative LLMs. These human experts have backgrounds in poetry studies or English literature, including professional poets, doctoral students, post-doc researchers, professors, and other researchers. We designed a prompting template (Li & Wang, 2024) for LLM-as-a-judge and a survey for human judges based on POEMetric, asking them to answer questions after reading the generation prompt and the poem written in response to the prompt. The questions comprised 10 multiple-choice questions (in line with the 10 metrics in POEMetric) using a 5-point Likert scale, asking the evaluators to score from 1 (Strongly Disagree) to 5 (Strongly Agree), and 3 open-ended questions where the evaluators could comment on why they gave that score in the previous question. The template of the survey for human experts and the evaluation prompt for the LLM judge can be found in Appendix C.

## 5 EXPERIMENTS

### 5.1 EXPERIMENT SET-UP

To better compare different LLMs, we adopted default sampling parameters for each LLM. For open-source models, we applied vLLM (Kwon et al., 2023) to deploy them on local GPUs. We guaranteed that each LLM received the same text prompt. As for system prompt, we adopted the default setting. In choosing LLM-as-a-judge, our pilot study (see Appendix D) showed that, Gemini-2.5-Pro, compared with DeepSeek-R1 and GPT-4o, yielded higher agreement with human experts (PAo=0.662 vs. 0.548/0.438) and superior discriminative ability in evaluating Overall Poem Quality (Std. Dev. 0.63 vs. 0.20/0.22), which were crucial for ensuring evaluation validity. At the same time, averaging with other LLMs would have introduced noise and bias. Thus, we chose Gemini-2.5-Pro (Team, 2025b) as the single LLM judge, which is also one of the strongest generalist LLMs across different benchmarks (Phan et al., 2025; Rein et al., 2024; AIM; Jain et al., 2024; Wei et al., 2024; Yue et al., 2024; Chiang et al., 2024) with free access for the research community.

## 5.2 LLM SELECTION AND POEM GENERATION

We prompted 30 models of 7 leading AI companies for poem generation; an overview of the selected models is shown in Appendix E. We employed a simple prompting template (see Figure 2) to include the form, rhyme, meter and theme of each human poem. Each LLM responded to 203 prompts generated based on the human poem dataset, totaling 6,090 LLM poems. A general description of the linguistic features of the human-LLM poem dateset, such as most frequent words, top opening words, and most common imagery, can be found in Appendix F.

## 6 RESULTS

In this section, with a focus on the best-performing LLMs representative of the 7 AI companies, we will first present a case study, discuss the results produced by rule-based evaluation, then turn to the evaluation by Gemini-2.5-Pro, and finally explain its similarity to human evaluation results.

### 6.1 A CASE STUDY: AN ILLUSTRATIVE COMPARISON

To provide a concrete illustration of the aggregate findings, we begin with a direct comparison. This case study showcases a poem generated by a high-performing LLM, i.e., DeepSeek-R1, alongside a human-written poem, based on the same prompt. In Figure 3, before revealing the authors, can the reader discern which poem was written by a human? By examining the works side-by-side, along with their POEMetric scores, the nuanced differences between machine-generated text and human artistry become tangibly clear. Our LLM judge, Gemini-2.5-Pro, decided that DeepSeek-R1's output was technically flawless, adhering strictly to the prompt's formal constraints, and employed evocative imagery and literary devices, leading to higher scores across various dimensions than the human-written work - do you agree with our judge? More showcases can be found in Appendix G.

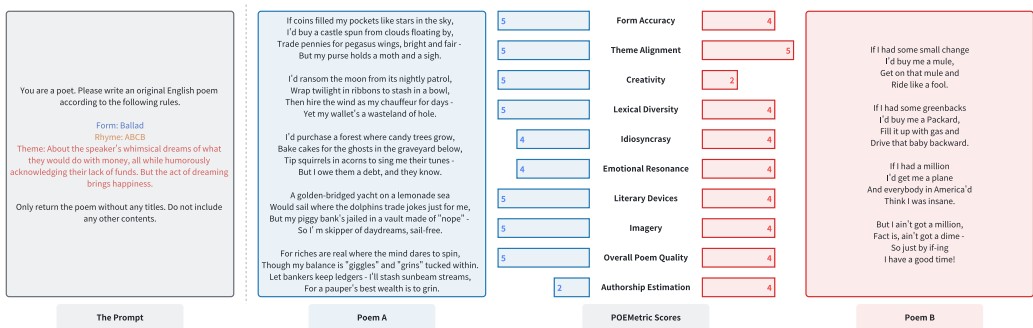

Figure 3: A showcase of the poems by DeepSeek-R1 (Poem A) and a human poet (Poem B) in response to the same prompt. The bar charts show their POEMetric scores judged by Gemini-2.5-Pro.

Furthermore, to illuminate the generative process of advanced models, we present the Chain-of-Thought (CoT) output from DeepSeek-R1 when generating its poem in Figure 4. This internal monologue reveals a structured, intentional process of creative reasoning, which demonstrates that the model's process is not a black box. It methodically deconstructs tasks, plans its structure, and even critiques its own word choices, which is very similar to the thinking process of a human poet.

### 6.2 RULE-BASED EVALUATION

Figure 5 shows the rule-based form accuracy, MATTR, and repetition rate of each author. First, our automated form detection algorithm, including the meter and rhyme patterns, examined the 7 representative LLMs, where Gemini-2.5-Pro and Claude-3.7-Sonnet showed relatively high form accuracy (0.50 and 0.47). Second, LLMs demonstrated higher lexical diversity than humans did according to MATTR. Last but not the least, LLM-generated poems exhibited high repetition rates on the word level when compared with the human poems, suggesting pronounced imitation of human works.

```
<think>
Alright, the user wants a ballad with ABCB rhyme scheme on the theme of whimsical dreams about money while being broke but happy. Let me break this down.
First, ballad structure usually has quatrains (four-line stanzas) with the second and fourth lines rhyming. ABCB means lines 2 and 4 rhyme each time. Need to make sure each stanza follows that.
Theme is whimsical dreams about money, humorous acknowledgment of lack of funds, and finding happiness in dreaming. So the poem should have imaginative ideas of what to do with money, contrast with reality, and end on a positive note about the joy of dreaming.
Let me brainstorm some whimsical ideas. Maybe buying a castle made of clouds, trading coins for the moon, a golden boat, or a forest of candy trees. These are playful and show extravagance without real money.
Now, contrast with the reality. Lines like "but my pockets are bare" or "my wallet's a wasteland" to show they can't afford these things. Mix in humor, like owing debts to a squirrel or a piggy bank jail.
Finally, the happiness from dreaming. End with stanzas emphasizing that even without money, dreaming brings joy and they're "rich in the tunes" of their dreams.
Check the meter. Ballads often have a rhythm, maybe alternating between 8 and 6 syllables. Let me count syllables for each line to ensure a flow. Not too strict, but a consistent beat.
Make sure each stanza follows ABCB. For example:
Line 1: dream idea
Line 2: action or detail (ends with rhyme B)
Line 3: reality check
Line 4: humorous line (rhymes with line 2)
Let me draft a few stanzas, check the rhymes and flow. Revise where necessary to keep the whimsical and humorous tone. Ensure the ending ties back to the theme of happiness from dreaming.
</think>
```

Figure 4: Chain-of-Thought (CoT) process from DeepSeek-R1 for the poem generation. The model explicitly breaks down the prompt, plans the thematic progression stanza by stanza, brainstorms mter and rhymes, and attempts to strategically insert literary devices.

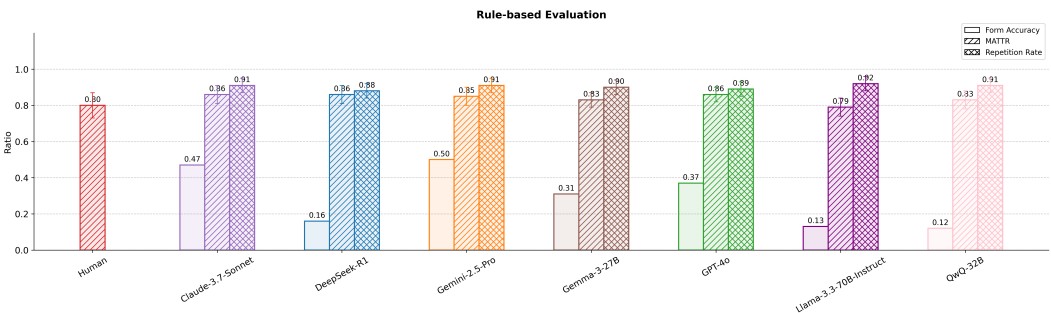

Figure 5: Rule-based evaluation results. LLMs were able to achieve high form accuracy and MATTR. However, their poems were highly repetitive compared with the original human poems.

## 6.3 LLM-AS-A-JUDGE EVALUATION

**Basic instruction-following abilities** In Figure 6, Gemini-2.5-Pro scored the highest in terms of **form accuracy** (4.26) and **theme alignment** (4.99), suggesting outstanding instruction-following abilities compared with the other LLMs, while Llama-3.3-70B-Instruct ranked low in both metrics (2.29, 4.91). We found that some of the poorly performing LLMs would stick to a default form for a certain type of poem; for instance, they would use the common rhyme pattern ABAB when writing ballads, instead of following the specific ABCB instruction in the prompt given, thus resulting in unsatisfying performance in form accuracy.

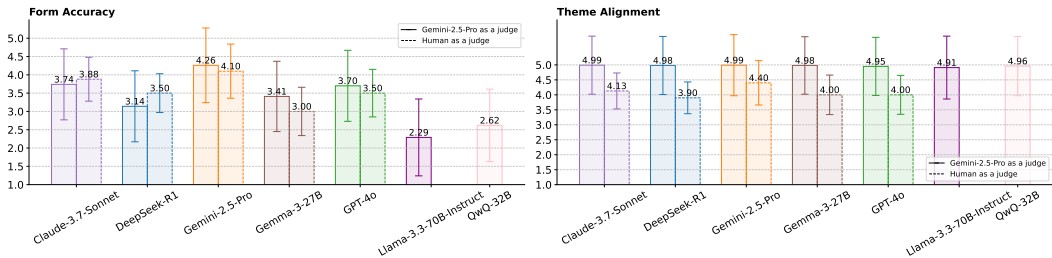

Figure 6: Form accuracy and theme alignment scores. Gemini-2.5-Pro achieved the highest scores in both dimensions, whereas Llama-3.3-70B-Instruct ranked the lowest among the 7 LLMs.

**Advanced creative abilities**   As shown in Figure 7, compared with LLM-generated poems, the poems written by humans excelled in terms of **creativity** (4.02), **idiosyncrasy** (3.95), **emotional resonance** (4.06), and the use of **imagery** (4.49) and **literary devices** (4.67). Among the 7 representative LLMs, DeepSeek-R1 yielded the best performance while Llama-3.3-70B-Instruct achieved the lowest scores. Meanwhile, as somewhat expected, LLMs showed significantly less idiosyncrasy in their poems, indicating a lack of personal distinctiveness or experience. However, DeepSeek-R1 (3.85) outperformed humans (3.82) in terms of lexical diversity.

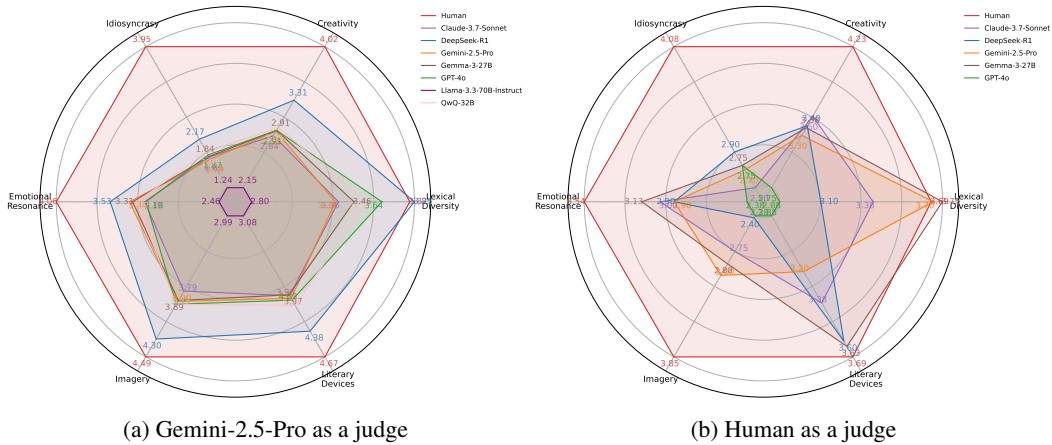

(a) Gemini-2.5-Pro as a judge          (b) Human as a judge

Figure 7: Advanced creative abilities. Compared with LLMs, human poets excelled in creativity, idiosyncrasy, emotional resonance, and use of imagery and literary devices.

**General appraisal**   Figure 8 demonstrates the overall poem quality and human authorship estimation of the poems. For one thing, poems written by humans achieved a higher mean score (4.22) than those generated by LLMs in terms of the **overall quality** due to the effective and idiosyncratic use of language by humans, according to the comments given by Gemini-2.5-Pro in the open-ended questions. Following humans was DeepSeek-R1, which was only slightly better than the other LLM authors. For another, although **authorship** was not revealed to Gemini-2.5-Pro as a judge, it was generally able to distinguish between a human poem and an LLM poem. Of all the 203 human poems, Gemini-2.5-Pro was able to recognize 80 poems (39.4%), either by reciting the original poem or by recognizing the distinctive style of a poet. Figure 9 shows the overall performance of humans and all 30 LLMs, and the scores were averaged across basic instruction-following abilities, advanced creative abilities, and poem quality. There was a general tendency that models with more parameters within the same family series performed better in poem generation. Thinking models were not necessarily better than their non-thinking family members; for instance, GPT-4o and GPT-4 ranked higher than o1 and o3-mini. Besides, DeepSeek-R1-Distill models were generally worse than the original models, except that Distill-Llama-3.3-70B performed better than its original. More results of humans and all 30 LLMs in each specific dimension can be found in Appendix H.

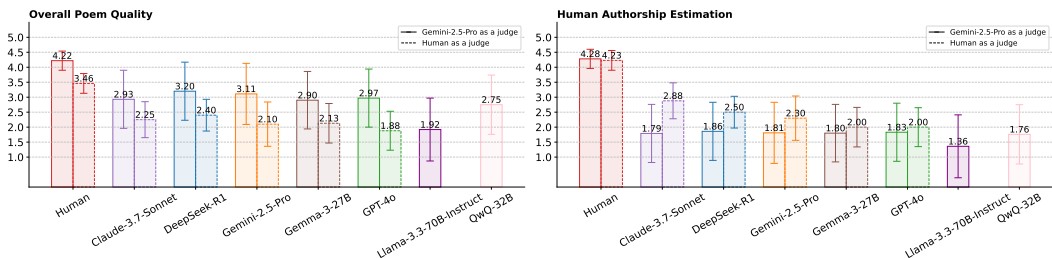

Figure 8: Overall poem quality and human authorship estimation scores. Humans ranked first in terms of overall poem quality, and human poems remained largely distinguishable from LLM poems.

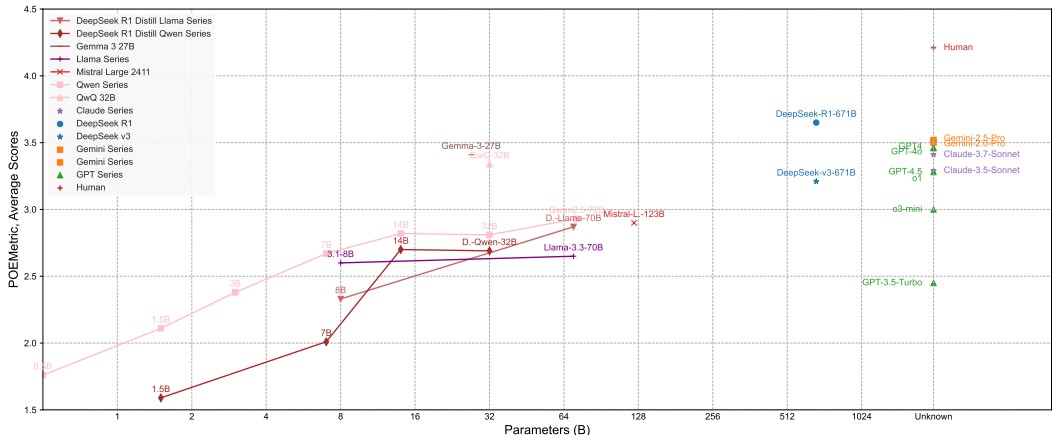

Figure 9: The mean scores of POEMetric of human poets and 30 LLMs, evaluated by Gemini-2.5-Pro. Models with more parameters within the same family generally performed better. Thinking models were not necessarily better than their non-thinking family members, as GPT-4o and GPT-4 ranked higher than o1 and o3-mini. DeepSeek-R1-Distill models were generally worse than the original models, except Distill-Llama-3.3-70B performed better than its original.

## 6.4 HUMAN VALIDATION

In order to validate the evaluation results given by Gemini-2.5-Pro, we calculated its Proportion Agreement, Observed (PAo) (Neuendorf, 2017) with the expert human evaluators in order to test inter-rater reliability. The Proportion Agreement, Observed (PAo) is calculated using the following formula:

$$PAo = \frac{2A}{n_A + n_B},$$

where $A$ is the number of agreements between the raters, $n_A$ is the total number of ratings by Rater A, and $n_B$ is the total number of ratings by Rater B. This formula quantifies the degree of agreement between two raters, providing a measure of how often their ratings coincide. The PAo test between the scores given by Gemini-2.5-Pro as a judge and the human evaluators across the 10 multiple-choice questions found strong agreement (0.662). In addition, we also calculated Cohen's Kappa and Spearman's rank correlation coefficient, where strong correlation was found in human-LLM agreement (Quadratic Weighted Kappa $\kappa = 0.361$, Spearman Correlation $\rho = 0.378$). This performance echoes existing studies involving LLM-human inter-rater agreement; for example, in evaluating an agentic reviewer Jiang & Ng (2026), $\rho$ between one human reviewer and another is 0.41, whereas $\rho$ between AI and one human reviewer is 0.42; in 20 NLP Evaluation Tasks Bavaresco et al. (2025), the agreement between the best-performing LLM and humans falls between $\kappa = 0.28\pm0.32$. Hence the results in our study were robust. In what follows, we will discuss the similarities and discrepancies between the results given by the two groups of judges.

As shown in Figure 6, there was a high similarity between human evaluators and Gemini-2.5-Pro in terms of the **form accuracy** and **theme alignment** of the LLM-generated poems, with Gemini-2.5-Pro and Claude-3.7-Sonnet ranking top. However, when judging theme alignment, human evaluators tended to give higher scores than Gemini-2.5-Pro did. As for **advanced creative abilities**, in Figure 7, both Gemini-2.5-Pro and the human judges decided that, compared with LLM poems, human poems excelled in terms of creativity, idiosyncrasy, emotional resonance, and the use of imagery and literary devices. By comparison, some LLMs were thought to be able to use a more varied vocabulary than humans did, though there was a disagreement in which LLM was more lexically diverse. In Figure 8, as for **overall poem quality**, it is shown that the human judges were more restrained in giving high scores: even the first-ranking human poems achieved only a mean score of 3.46, meaning the human evaluators were prone to agree that these were good poems, but not so certain. As for estimating **authorship**, human evaluators were also generally able to tell if a poem was written by a

human or an LLM. Compared with Gemini-2.5-Pro's relatively high ratio of recognizing the original human poems (39.4%), within the 13 human poems evaluated by human experts, only 1 poem was recognized as a famous poem, and yet all 13 poems were scored 3 (neutral) or higher (agreeing or strongly agreeing this poem was written by a human). This implies that, though the human judges could not recognize as many original poems as Gemini-2.5-Pro could, they were still likely to find out the authorship of a poem. Nevertheless, in the face of LLM-generated poems, the human evaluators were less confident about their authorship compared with Gemini-2.5-Pro as a judge.

## 7 CONCLUSION AND LIMITATIONS

We introduce POEMetric, the most comprehensive evaluation framework for poetry generation so far. We also curated a human poem dataset, covering 203 poems of different poetic forms and themes, and experimented with 30 state-of-the-art LLMs. Although the top models have the capabilities of writing poems of certain styles and themes, they still fall short of attaining advanced creative abilities such as creativity, idiosyncrasy, evoking emotional resonance, and skillful use of imagery and literary devices. Moreover, our findings have demonstrated the effectiveness and efficiency of automated poetry evaluation with POEMetric. More explorations are encouraged to adjust POEMetric to evaluate free-style poems. Besides, this paper only examines the English language, while POEMetric is applicable to other low-resource languages as well. We leave it to future work.

## 8 ETHICS STATEMENT

The research presented in this paper adheres to the ICLR Code of Ethics. In our commitment to scientific excellence and transparency, we introduce POEMetric as a comprehensive framework and provide our code and dataset in the supplementary materials to foster reproducible research, with our methods, model selection, and limitations detailed throughout the paper. All work involving human participants was conducted under Institutional Review Board (IRB) approval, with informed consent and full data anonymization to protect privacy and honor confidentiality. Furthermore, we respect intellectual property by sourcing our dataset from properly credited, publicly accessible archives and building upon prior research as detailed in Section 2.

## 9 REPRODUCIBILITY STATEMENT

To facilitate the full reproducibility of our findings, we have made all key components of our research publicly available. The evaluation framework and methodology are clearly presented in Section 4. The code for our rule-based evaluation algorithm and the curated human poem dataset are provided anonymously in the supplementary materials. Comprehensive details regarding our experimental setup, including the complete list of the 30 LLMs evaluated (Appendix E), the model configurations (Section 5), and the precise generation prompt template (Figure 2), are provided to enable the replication of our poem generation process. Furthermore, the exact evaluation prompt used for the LLM-as-a-judge and the full survey administered to our human experts are included in Appendix C, ensuring that our multi-faceted evaluation can be independently verified and extended.

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

## A    FIXED FORMS OF ENGLISH POETRY

**Ballad**    Ballads are usually long poems consisting of quatrains (4-line stanzas, where a stanza means a section of a poem), following the rhyme pattern of ABCB or ABAB for each stanza. The two main types of ballads, the traditional folk ballads and the literary ballads, adopt varied meter patterns, and sometimes creative forms such as 6-line or 8-line stanzas with new rhyme patterns.

**Ghazal**    Originating in the Arabic poetry, ghazals are a set of couplets (2-line stanzas). Each couplet ends on the same word or phrase (the *radif* ), and is preceded by the couplet's rhyming word (the *qafia*, which appears twice in the first couplet).

**Limerick**    A traditional limerick is a stanza of 5 lines, with a fixed rhyme pattern of AABBA and varying meter patterns for each line. Later limericks were popularized by the poet Edward Lear, which consist of a 4-line stanza rhyming AABA, with the third line comprising two sentences split by a comma and both rhyming B.

**Pantoum**    The pantoum is a Malay verse form, a series of quatrains with the second and fourth lines of each quatrain repeated as the first and third lines of the next. The second and fourth lines of the final stanza repeat the first and third lines of the first stanza.

**Sestina**    The sestinas are a complex French verse form, usually unrhymed, consisting of six stanzas of six lines each and a three-line envoi. The end words of the first stanza are repeated in a different order as end words in each of the subsequent five stanzas; the closing envoi contains all six words, two per line, placed in the middle and at the end of the three lines.

**Sonnet**    Sonnets usually consist of 14 lines following the meter pattern of iambic pentameter, which is a line of verse composed of ten syllables arranged in five metrical feet (iambs), each of which consists of an unstressed syllable followed by a stressed syllable. There are different types of sonnets. The Petrarchan subcategory usually consists of one octave (8-line stanza) and one sestet (6-line stanza), and adopts a typical rhyme pattern of ABBAABBA CDCDCD/CDECDE. An English variation of the Petrarchan sonnets, i.e., the Italian sonnets, rhyme with ABBAABBA CDDCEE. The Shakespearean and Spenserian types usually comprise three quatrains followed by a couplet, each with different rhyme patterns. Apart from these, poets have also created new patterns such as 16-line sonnets and reversed sonnets.

**Villanelle**    As a French verse form, a villanelle consists of five three-line stanzas and a final quatrain, with the first and third lines of the first stanza repeating alternately in the following stanzas and forming the final couplet in the quatrain.

## B  THE ALGORITHM OF RULE-BASED FORM ACCURACY

In detecting the form accuracy via a rule-based algorithm, we tested on a subset of the human poems to optimize the trade-off between precision and recall for form detection. A higher threshold would sort out almost perfect poems, but would incorrectly reject many poems that contain minor stylistic variations, thus unfairly penalizing creativity. A lower threshold would be too lenient; it would incorrectly accept many poorly-formed poems. Therefore, we opted for a 0.7 threshold. The algorithm is as follows.

---

**Algorithm 1** Rule-Based Poetry Form Evaluation (POEMetric)

---

**Input**: Poem text $T$, Target Form $F$, Target Meter $M$, Target Rhyme $R$
**Output**: Boolean indicating if $T$ satisfies the constraints

1:  **Phase 1: Linguistic Feature Extraction**
2:  $W \leftarrow$ Tokenize$(T)$                       ▷ Split into lines and words, remove punctuation
3:  $A_{meter} \leftarrow \emptyset$, $A_{rhyme} \leftarrow \emptyset$
4:  **for** each line $l \in W$ **do**
5:      $P_{stress} \leftarrow$ ExtractStress$(l, \text{CMUdict})$                 ▷ Map to 'u', 'S', '*'
6:      $A_{meter}$.append$(P_{stress})$
7:      $w_{end} \leftarrow$ LastWord$(l)$
8:      $P_{phoneme} \leftarrow$ ExtractRhymeFoot$(w_{end}, \text{CMUdict})$
9:      $A_{rhyme}$.append$(P_{phoneme})$
10: **end for**
11: $S_{rhyme} \leftarrow$ MapToRhymeScheme$(A_{rhyme})$              ▷ e.g., 'A', 'B', 'A', 'B'
12: **Phase 2: Form-Specific Structural Validation**
13: $\mathcal{F}_{fixed} \leftarrow$ {Limerick, Villanelle, Sestina, Pantoum, Ghazal}
14: **if** $F \in \mathcal{F}_{fixed}$ **then**
15:     $isValid \leftarrow$ False
16:     **if** $F =$ Ghazal **then**
17:         $isValid \leftarrow$ CheckCouplets$(W) \wedge$ CheckRadifQafia$(W)$
18:     **else if** $F =$ Sestina **then**
19:         $isValid \leftarrow$ CheckPermutations$(W) \wedge$ CheckEnvoi$(W)$
20:     **else if** $F =$ Villanelle **then**
21:         $isValid \leftarrow$ CheckRefrains$(W) \wedge$ CheckGlobalRhyme$(S_{rhyme})$
22:     **else if** $F =$ Pantoum **then**
23:         $isValid \leftarrow$ CheckLineRepetitions$(W)$
24:     **else if** $F =$ Limerick **then**
25:         $isValid \leftarrow$ CheckPattern$(S_{rhyme}, \text{'AABBA'} \vee \text{'AABA'})$
26:     **end if**
27:     **if not** $isValid$ **then return** False
28:     **end if**
29:     $R \leftarrow$ None                       ▷ Bypass general rhyme check for fixed forms
30: **end if**
31: **Phase 3: Meter and Rhyme Validation (with Tolerance)**
32: **if** $M \neq$ None **then**
33:     $E_{meter} \leftarrow$ GetExpectedPattern$(M)$
34:     $MatchRatio \leftarrow \frac{1}{|A_{meter}|} \sum_{p \in A_{meter}} \mathbb{I}(p \approx E_{meter})$
35:     **if** $MatchRatio < 0.7$ **then return** False
36:     **end if**                            ▷ 70% threshold
37: **end if**
38: **if** $R \neq$ None **then**
39:     $MatchRatio \leftarrow$ CalculateRhymeMatch$(S_{rhyme}, R)$
40:     **if** $MatchRatio < 0.7$ **then return** False
41:     **end if**                            ▷ 70% threshold
42: **end if**
43: **return** True

---

## C    THE POEMetric-BASED LLM PROMPT AND HUMAN SURVEY

Below are the prompting template for LLM-as-a-judge, and the survey template for human expert judges, which share the same set of POEMetric-based questions.

**POEMetric-based Human Expert Survey**

Please read the prompt (i.e. the instructions that were given for writing each of the specific poems reproduced below), and then read the poem that was written in response to this prompt. Then answer the questions that follow. Please do not check any online or offline resources while completing this survey; we are interested in your own direct and personal response!

We will give you 10 poems in total; please do as many as you can. Do not worry if you could not finish all of them.

### Poem 1

### The prompt

{generation_prompt}

### The poem

{the_poem}

### Questions

Below are 10 required multiple-choice questions and 3 optional open-ended questions. For each multiple-choice question, please score the poem from 1 to 5 based on how strongly you agree or disagree with the question.

1. The poem follows the given prompt in terms of form, including meter and rhyme where applicable. (Required) ____________

 1 - Strongly disagree

 2 - Disagree

 3 - Neutral

 4 - Agree

 5 - Strongly agree

2. The poem follows the given prompt in terms of its theme. (Required) ____________

 1 - Strongly disagree

 2 - Disagree

 3 - Neutral

4 - Agree

5 - Strongly agree

3. The poem uses a varied vocabulary. (Required) __________

 1 - Strongly disagree

 2 - Disagree

 3 - Neutral

 4 - Agree

 5 - Strongly agree

4. The poem is a creative work. (Required) __________

 1 - Strongly disagree

 2 - Disagree

 3 - Neutral

 4 - Agree

 5 - Strongly agree

5. This poem shows idiosyncrasy. (Required) __________

 1 - Strongly disagree

 2 - Disagree

 3 - Neutral

 4 - Agree

 5 - Strongly agree

6. This poem evokes emotional resonance. (Required) __________

 1 - Strongly disagree

 2 - Disagree

 3 - Neutral

 4 - Agree

 5 - Strongly agree

7. The imagery in this poem is used well. (Required) __________

0 - N/A (No imagery is used)

1 - Strongly disagree

2 - Disagree

3 - Neutral

4 - Agree

5 - Strongly agree

8. At least one of the literary devices listed below is used well in the poem. (Required) __________

- Simile
- Metaphor
- Personification
- Allusion

0 - N/A (No literary devices are used)

1 - Strongly disagree

2 - Disagree

3 - Neutral

4 - Agree

5 - Strongly agree

9. Please comment on why you gave the answer that you did for question 8 above.

_______________________________________________________________

10. This is a good poem. (Required) __________

1 - Strongly disagree

2 - Disagree

3 - Neutral

4 - Agree

5 - Strongly agree

11. Please comment on why you gave the answer that you did for question 10 above.

_______________________________________________________________

12. The poem is written by a human. (Required) ___________

 1 - Strongly disagree

 2 - Disagree

 3 - Neutral

 4 - Agree

 5 - Strongly agree

13. Please give comments on why you gave the answer that you did for question 12 above.

_______________________________________________________________________

**POEMetric-based LLM Evaluation prompt**

# Role Description
You are a professional poetry critic and analyst. Your job is to evaluate English poetry written by human beings and English poetry generated by large language models.

# Task Definition
I will ask you to evaluate one poem by answering 10 multiple-choice questions and 3 open-ended questions, without telling you if the poem is written by a human or an LLM. For each multiple-choice question, please score the poem from 1 to 5 based on how strongly you agree or disagree with the question.

# Task Procedures
- Step 1, read the prompt (i.e. the instructions that were given for writing each of the specific poems reproduced below).
- Step 2, read the poem that was written in response to this prompt.
- Step 3, answer the questions that follow.

# Context

## The prompt
{generation_prompt}

## The poem
{the_poem}

## Questions

1. The poem follows the given prompt in terms of form, including meter and rhyme where applicable.
1 - Strongly disagree
2 - Disagree
3 - Neutral
4 - Agree
5 - Strongly agree

2. The poem follows the given prompt in terms of its theme.
1 - Strongly disagree
2 - Disagree
3 - Neutral
4 - Agree
5 - Strongly agree

3. The poem uses a varied vocabulary.
1 - Strongly disagree
2 - Disagree
3 - Neutral
4 - Agree
5 - Strongly agree

4. The poem is a creative work.
1 - Strongly disagree
2 - Disagree
3 - Neutral
4 - Agree
5 - Strongly agree

5. This poem shows idiosyncrasy.
1 - Strongly disagree
2 - Disagree
3 - Neutral
4 - Agree
5 - Strongly agree

6. This poem evokes emotional resonance.
1 - Strongly disagree
2 - Disagree
3 - Neutral
4 - Agree
5 - Strongly agree

7. The imagery in this poem is used well.
0 - N/A (No imagery is used)
1 - Strongly disagree
2 - Disagree
3 - Neutral
4 - Agree
5 - Strongly agree

8. At least one of the literary devices listed below is used well in the poem.
- Simile
- Metaphor
- Personification
- Allusion
0 - N/A (No literary devices are used)
1 - Strongly disagree

2 - Disagree
3 - Neutral
4 - Agree
5 - Strongly agree

9. Please comment on why you gave the answer that you did for question 8 above.

10. This is a good poem.
1 - Strongly disagree
2 - Disagree
3 - Neutral
4 - Agree
5 - Strongly agree

11. Please comment on why you gave the answer that you did for question 10 above.

12. The poem is written by a human.
1 - Strongly disagree
2 - Disagree
3 - Neutral
4 - Agree
5 - Strongly agree

13. Please give comments on why you gave the answer that you did for question 12 above.

## Output Format
For each multiple-choice question, please give your score directly, without any explanation. Your output should be in the json format as follows:

{"1": <insert your score here>, "2": <insert your score here>, ..., "9": "<insert your comments here>", ...}

# D LLM-AS-A-JUDGE JUSTIFICATION

We agree that cross-model validation is the ideal, and we performed this analysis in our pilot studies. Our results revealed that averaging scores from multiple LLMs would degrade the evaluation quality, as other leading models proved to be flawed evaluators in two key ways:

**Low Agreement with Human Experts**  We tested DeepSeek-R1 and GPT-4o, and they demonstrated substantially lower inter-rater reliability with our human experts. The Observed Proportion Agreement (PAo) (Neuendorf, 2017) was low for GPT-4o (0.548) and DeepSeek-R1 (0.438), but strong for Gemini-2.5-Pro (0.662). This divergence from human consensus would introduce significant noise and undermine the validity of our findings.

**Lack of Discrimination Ability**  Other models failed to distinguish between high- and low-quality poems in the "overall poem quality" dimension. As shown in Table 2, the extremely low standard deviations for DeepSeek-R1 (0.20) and GPT-4o (0.22) confirm that their scores were tightly clustered at the high end of the scale (as shown by their mean scores of 4.26 and 3.69). Including them would inevitably introduce bias and noise. In contrast, the standard deviation of Gemini-2.5-Pro's scores (0.63) was much closer to that of our human experts (1.09), indicating it was a far more reliable and discerning instrument for measurement.

Table 2: Human vs LLM-as-a-judge evaluation results on the "overall poem quality" dimension.

| Judge | Human | DeepSeek-R1 | Gemini-2.5-Pro | GPT-4o |
|---|---|---|---|---|
| Mean | 2.43 | **4.26** | 3.00 | 3.69 |
| Standard Deviation | **1.09** | 0.20 | 0.63 | 0.22 |

In conclusion, our selection of Gemini-2.5-Pro was a rigorous decision to ensure the quality and validity of our evaluation.

# E An overview of the 30 selected LLMs

Table 3: The features of 30 selected LLMs.

|  | **Non-Thinking** | **Thinking** |
| --- | --- | --- |
| **Open-source** | DeepSeek-v3 (Liu et al., 2024) DeepSeek-R1-Distill-Llama-8B/70B (Guo et al., 2025) DeepSeek-R1-Distill-Qwen-1.5B/7B/14B/32B (Guo et al., 2025) Gemma-3-27B (Team et al., 2025) Llama-3.1-8B/3.3-70B-Instruct (Grattafiori et al., 2024) Mistral-Large-2411-123B (Team, 2024) Qwen2.5-0.5B/1.5B/3B/7B/14B/32B/72B-Instruct (Qwen et al., 2025) | DeepSeek-R1 (Guo et al., 2025) QwQ-32B (Team, 2025e) |
| **Closed-source** | Claude-3.5-Sonnet (Anthropic, 2024) Gemini-2.0-Pro (Team, 2025a) GPT-3.5-Turbo (Team, 2023) GPT-4 (Achiam et al., 2023) GPT-4o (Hurst et al., 2024) GPT-4.5 (Team, 2025c) | Claude-3.7-Sonnet (Anthropic, 2024) Gemini-2.5-Pro (Team, 2025b) o1 (Jaech et al., 2024) o3-mini (Team, 2025d) |

# F  LINGUISTIC FEATURES OF THE HUMAN-LLM POEM DATASET

Figure 10 demonstrates the top 20 case-insensitive words in the human poems and the poems generated by 7 state-of-the-art LLMs representative of the 7 AI companies, with stop words removed. Among them, Claude 3.7 Sonnet resembles humans the most, with cosine similarity of 0.602. Figure 11 illustrates the most frequent opening words and imagery used by LLMs and human poets. For the choice of first words, each LLM has a distinctive taste. For example, Llama-3.3-70B-Instruct uses "In" significantly more than the other authors. Similarly, "The" appears more in poems generated by Gemma-3-27B and Gemini-2.5-Pro, while GPT-4o uses "Beneath" and Claude-3.7-Sonnet adopts "In" as the most common opening word. In comparison, human poets show a more balanced preference for choosing opening words. As for imagery, both LLMs and human poets tend to use the imagery "eyes", "sun" and "face", but each author also shows different preferences. While human poets frequently write about "water" and "god", DeepSeek R1 prefers "threads" and "bloom", QwQ-32B loves depicting "thread".

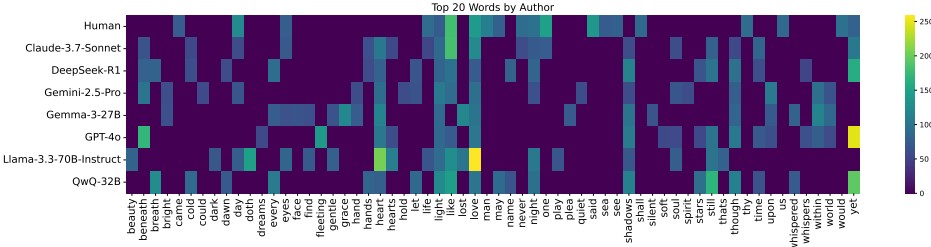

Figure 10: The top 20 words across the human and LLM poem datasets.

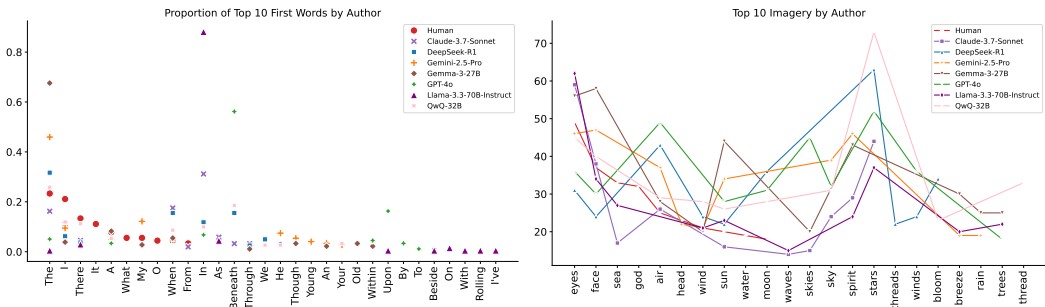

Figure 11: The top opening words and top imagery cross the human and LLM poem datasets..

# G    MORE SHOWCASES OF LLM AND HUMAN POEMS

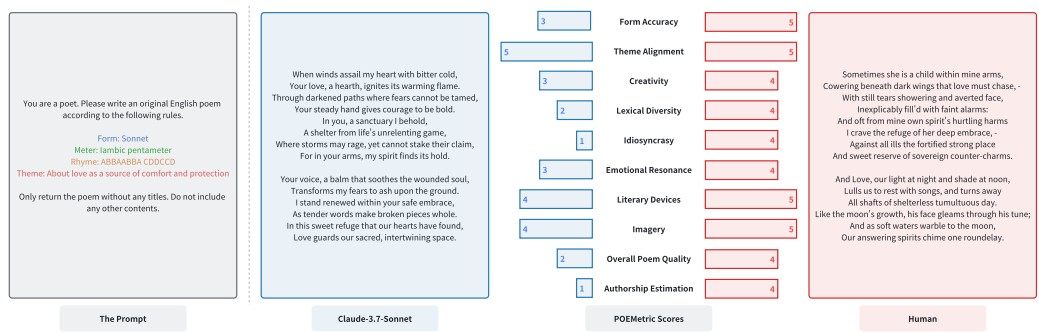

Figure 12: A showcase of the poems by Claude-3.7-Sonnet and a human poet in response to the same prompt. The bar charts show their POEMetric scores judged by Gemini-2.5-Pro.

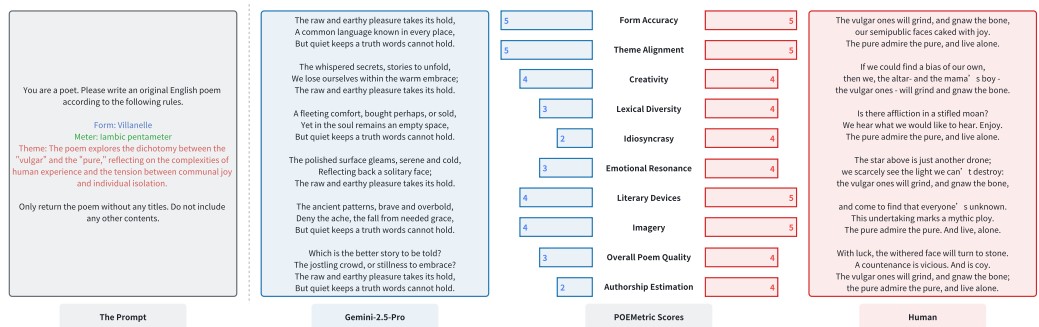

Figure 13: A showcase of the poems by Gemini-2.5-Pro and a human poet in response to the same prompt. The bar charts show their POEMetric scores judged by Gemini-2.5-Pro.

## H    POEMETRIC SCORES OF HUMAN POETS AND ALL 30 LLMS

The average scores of basic instruction-following abilities of all 30 LLMs are shown in Figure 14, the average scores of advanced creative abilities of both human poets and LLMs in Figure 15, those of overall poem quality in Figure 16, and those of human authorship estimation in Figure 17. Overall, models with more parameters within the same family series performed better in poem generation. Thinking models were not necessarily better than their non-thinking family members; for instance, GPT-4o and GPT-4 ranked higher than o1 and o3-mini. Besides, DeepSeek-R1-Distill models were generally worse than the original models, except that Distill-Llama-3.3-70B performed better than its original.

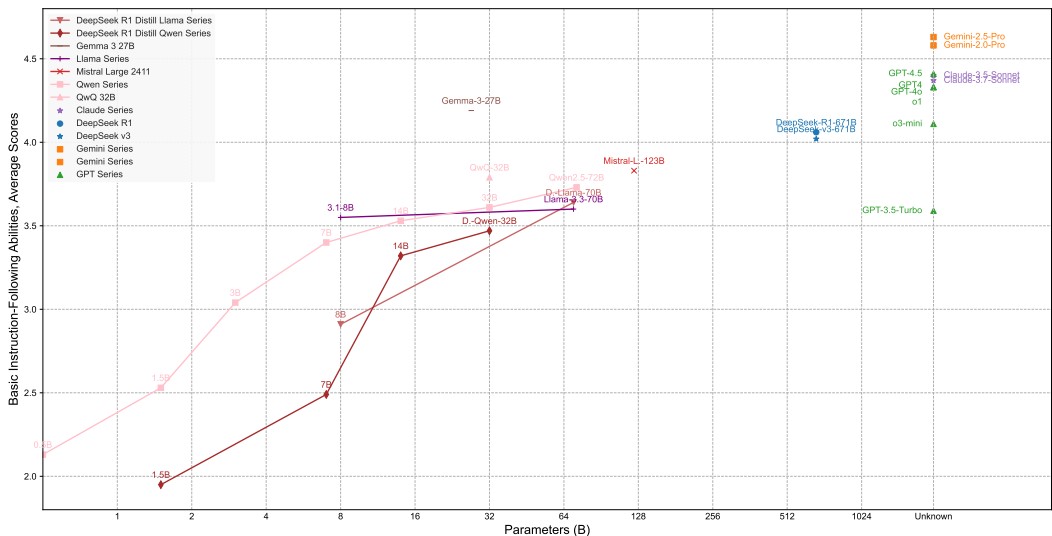

Figure 14: Basic Instruction-Following Abilities, Average Scores

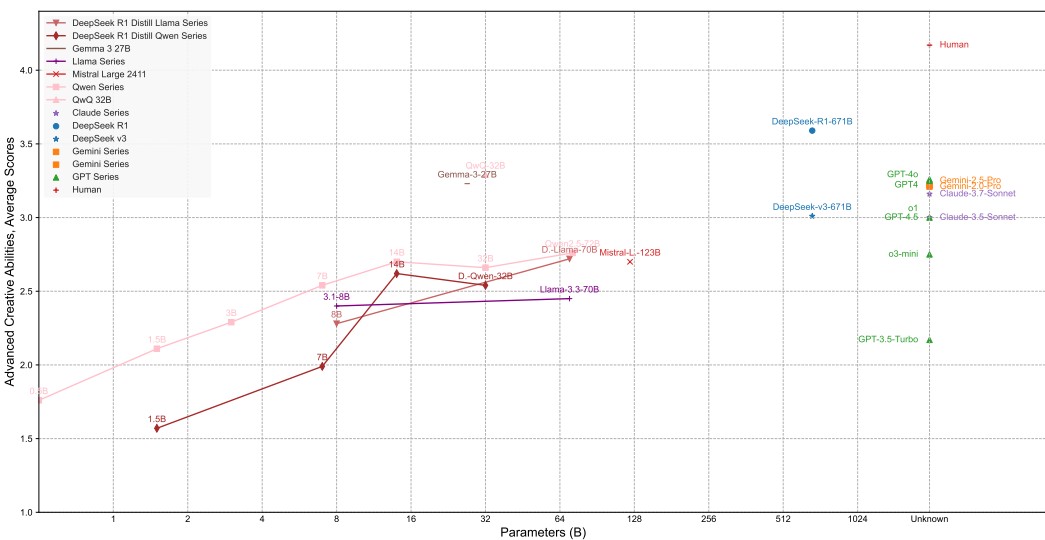

Figure 15: Advanced Creative Abilities, Average Scores

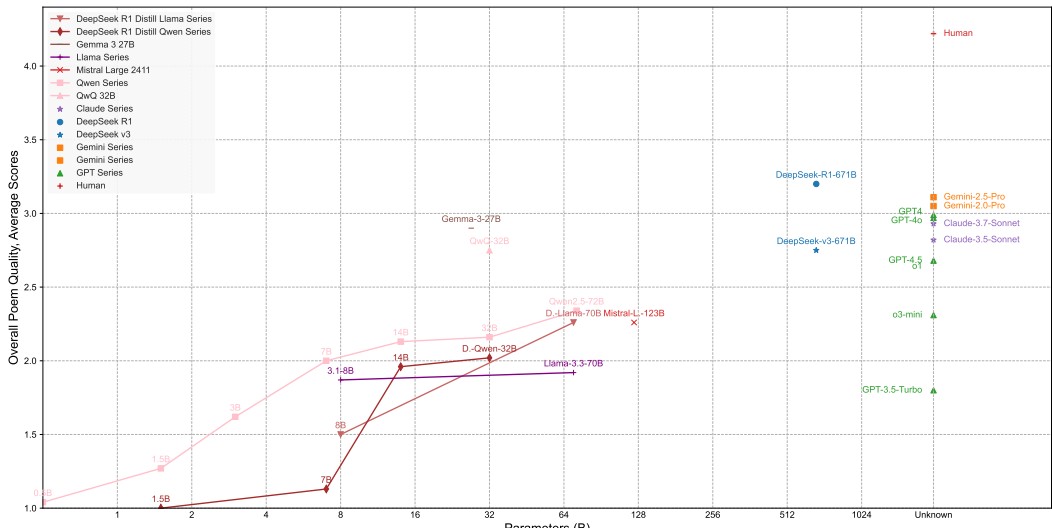

Figure 16: Overall Poem Quality, Average Scores

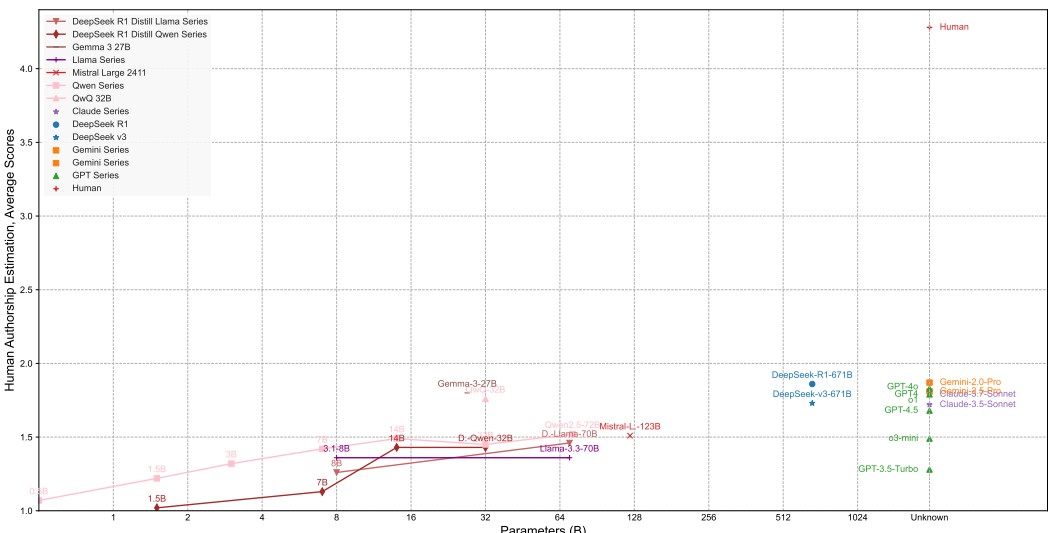

Figure 17: Human Authorship Estimation, Average Scores

## I   LLM USAGE STATEMENT

We have used LLMs only to aid or polish writing when drafting this paper.

