# OpenReview forum: "POEMetric: The Last Stanza of Humanity"
_ICLR.cc/2026/Conference — ICLR 2026 Poster_

### Official Review · Reviewer_xKLP · 2025-10-20

**Soundness:** 3
**Presentation:** 3
**Contribution:** 3
**Rating:** 6
**Confidence:** 4

**Summary:**

The authors introduce a novel evaluation dataset and framework for formal verse poetry. They first collect 203 human-created poems which they annotate with meter, rhyme pattern, topic, and other styles. These annotations are then used to condition a wide range of large language models to generate style-conditioned poetry. To evaluate generated (and to a lesser degree also human) poems, the authors propose various evaluation schemes. E.g., rule-based evaluation of form, meter, and rhyme scheme; and LLM/human-as-a-judge evaluation of basic instruction-following abilities, advanced creative abilities, and general appraisal. The authors show that LLMs-as-a-judge correlate well with human assessments and that generated poetry generally follows stylistic constraints well but fails to reach the same mastery of language (use of literary devices, creativty, emotional resonance, ...) that humans achieve.

**Strengths:**

* Poetry evaluation is an challenging research problem, especially for general-purpose large language models whose poetry generation performance is not well understood. The authors successfully investigate a gap and their findings will be useful for future research.
* The authors promise to release their dataset artifacts which will be useful for future work.
* The authors evaluate many different dimensions of poetry which give a broad overview of performance.

**Weaknesses:**

* The scope of the paper is a bit limited. It would have been interesting to see how some of the uncovered gaps between humans and generated poems could be closed either by model training or more advanced prompting. At least some suggestions for future work would have been insightful.

**Questions:**

* l.259 why would averaging with other LLMs introduce noise and bias? Wouldn't that just be model ensembling which is a common technique that usually works well? ByGPT5 (one of your cited papers) showed that general purpose LLMs (as of 2023) are bad at following stylistic constraints such as rhyme and meter. Do the authors have an intuition what changed with current LLMs which apparently now work much better?

Comments
* The poems in Figure 3 are too small to read comfortably.
* l480: ALthough -> Although

---

> ### Author Response · Authors · 2025-11-30
>
> We thank the reviewer for the positive assessment and for recognizing the value of our dataset and framework. We appreciate the insightful questions regarding model evolution and evaluation methodology.
>
> **1. Limited scope and future work suggestions (closing the gap).**
>
> We agree that bridging the gap between human and machine creativity is the natural next step. Based on our findings, we will discuss future directions in the final version:
>
> - **POEMetric as a Reward Model:**
> The automated components of our framework can serve as reward functions in RLHF (Reinforcement Learning from Human Feedback) pipelines to fine-tune models specifically for poetic nuance, moving beyond generic instruction following.
>
> - **Prompting and Chain-of-Thought (CoT):**
> Our case study of DeepSeek-R1’s "thinking process" suggests that explicit planning helps. Future work could explore more sophisticated prompting strategies and CoT prompting (e.g., "Plan the metaphor first, then write the stanza") to improve the Imagery and Emotional Resonance scores, where models currently lag behind humans.
>
> **2. Why would averaging with other LLMs introduce noise/bias? Isn't ensembling better?**
>
> While model ensembling is generally a robust technique, it relies on the assumption that the individual models are relatively reliable or have uncorrelated errors. In our pilot study (detailed in Appendix D), this assumption did not hold:
>
> -**Lack of Discrimination**
> As shown in Table 2, Appendix D, other candidate judges like GPT-4o and DeepSeek-R1 exhibited extremely low standard deviations (0.22 and 0.20) compared to humans (1.09). They tended to rate almost all poems as "excellent," failing to distinguish between high-quality and mediocre outputs.
>
> - **Low Agreement**
> Their agreement with human experts was significantly lower (PAo ~0.4-0.5) compared to Gemini-2.5-Pro (PAo 0.662).
>
> Therefore, averaging a "discriminative" judge (Gemini) with "non-discriminative/generous" judges would have diluted the signal, pulling the evaluation towards a compressed, high-score distribution that masks the very differences we aimed to measure. We chose quality over quantity for the judge selection.
>
> **3. Intuition on why current LLMs are much better at constraints (rhyme/meter) than 2023 models (ByGPT5)?**
>
> We attribute this performance leap to three key factors evolved since 2023:
>
> - **Structural Generalization from Code/JSON Training**
> Unlike natural language, which is permissive, code and structured data formats (like JSON) enforce rigid syntactic constraints and long-range dependencies (e.g., a closing brace } must match an opening brace { hundreds of tokens earlier). We hypothesize that extensive pre-training on strictly formatted data has endowed models with superior "structural attention". Writing a Sonnet (14 lines, fixed meter) or a Villanelle (repeating lines) is structurally analogous to generating valid JSON: the model must adhere to a pre-defined schema while filling in content. The capability to strictly follow code syntax transfers to the strict adherence to poetic forms.
>
> - **Instruction Tuning (RLHF)**
> Modern RLHF specifically penalizes constraint violation. Constraints are no longer treated as "style suggestions" (as in 2023) but as "hard instructions" akin to executable commands.
>
>
> **Other comments**
>
> **1. The size of Figure 3**: We apologize for the size of Figure 9 due to space limit. We would like to clarify that the current figure is a vector graphic, which allows for high-quality zooming.  In the revised manuscript, we will increase the font size and adjust the layout for better readability.
>
> **2. Typo**: We have corrected "ALthough" to "Although" (Line 480).

---

### Official Review · Reviewer_cK2s · 2025-10-27

**Soundness:** 3
**Presentation:** 4
**Contribution:** 2
**Rating:** 4
**Confidence:** 3

**Summary:**

This paper presents a method for poetry generation and evaluation using modern LLMs. The authors curate a new human dataset with rich metadata (title, topic, style, meter, etc) and use this to show that machine generated poetry is high quality in terms of the structural elements of poetry, but falls short on creativity, style and metaphor etc.

Overall this is well motivated and executed, however I feel that the main contributions of the work are the very detailed user study, as the method itself is quite simplistic, and as such this isn't a good fit for ICLR.

**Strengths:**

This paper’s main contribution was a detailed analysis of poetry outputs, and a careful and detailed comparison between various LLMs versus humans.

The dataset and evaluation should be useful to others working in the area, the wide range of poem types and rich metadata would make it a valuable resource.

The method used for generating and evaluating poetry was well motivated, clearly presented and sound (although many details were pushed to appendices and attached code, the latter which I haven't inspected.)

**Weaknesses:**

As stated above, I don’t think ICLR is the right venue to properly assess the human elements of the work.

The language generation and evaluation aspects were solid, but this is more a data + prompting exercise than a scientific contribution. The real contributions of the paper were the detailed evaluation, which is reflected in the way the paper in written, and the split of materials between the main text and the appendices.

Accordingly, I think this work would be better suited to other venues, e.g., computational linguistics, even CHI or more user-centric fields, where the reviewing process would be better able to judge the quality of the human study side of this work.

**Questions:**

1. There’s a lot of algorithmic detail in the appendix, around how the dataset is created that isn’t fleshed out and reproducible (appendix B, page 16, most steps are loaded: Levenshtein distance with walk? how to check for repetition and consistency? what is align?). (Note code is provided separately, according to the statement, so probably ok.)
1. The use of prompting seems a bit naive, as this is formatted as a user survey which may not be well handled by many LLMs. I expect that tailoring the prompt may improve the quality and robustness of model outputs. Consider k-shot prompting, providing examples of good vs bad outputs for each criteria, asking for ratings and rationales, versus ratings alone, etc.
1. Is thinking on/off a factor? There was a brief comparison of pairs of models with and without thinking, but I think this was confounded by model size. For some models thinking can be turned on and off. The same holds true for search, which may be relevant to evaluation.
1. Are the eval metrics closely correlated to one another? Perhaps you might distill down to a handful of metrics instead.
1. Might the model be directed to generate more creative poetry with more detailed prompting, or an iterative approach where feedback is given to make the eval more human-like (e.g., RL train over key evaluation metric)? This would be beyond the scope of one paper, but would be worthy of discussion.
1. The human expert survey for eval includes “Please do not check any online or offline resources while completing this survey; we are interested in your own direct and personal response!” – what about plagiarism of text and ideas from other sources? I note that there was no such directive given to the LLM as judge eval, is search thought to be useful here?
1. Agreement between human experts - the PoA analysis just uses LLM vs human setting, not human vs human. It's common to hold-out one human and compare them against the remaining humans as an upper bound.
1. Regarding confidence in predictions: LLMs are often overconfident, see the extensive work on calibration of LLMs. The discussion appears to rely on what is a rather unreliable signal.

---

> ### Author Response · Authors · 2025-11-30
>
> **1. Relevance to the ICLR Community**
>
> We thank the reviewer for acknowledging that the work is "well motivated and executed." However, we respectfully disagree with the characterization of this work as primarily a "user study" or a simple prompting exercise. We believe this view overlooks the core scientific contributions of the paper, which are squarely aligned with ICLR’s focus on evaluation methodologies, datasets, and understanding the capabilities of Generative AI.
>
> **1.1. Clarification of Contributions: A Benchmark, Not Just a Study**
>
> The human expert evaluation in our paper is not the end product; rather, it is the validation mechanism used to prove the reliability of our proposed framework, POEMetric. Much like in image generation or style transfer research, human evaluation is employed to validate a proposed metric (e.g., validating that a new metric correlates with human perception).
>
> Core Contribution: Our primary contribution is the POEMetric Framework itself, which includes:
> - A curated, annotated dataset of 203 poems with reverse-engineered constraints (Prompt-Poem pairs), filling a gap in "Instruction Following" datasets for creativity.
> - A Rule-Based Algorithm (validated with 0.804 mAP, see General Response) for objective form detection.
> - A Validated LLM-as-a-Judge Protocol (validated with 0.81 Ranking mAP against humans) for subjective evaluation.
>
> This framework provides the AI community with a reproducible, standardized yardstick to measure "Advanced Creative Abilities," going far beyond a one-off user study.
>
> **1.2. Relevance to the ICLR Community**
>
> ICLR has a strong tradition of publishing benchmarks and evaluation frameworks (e.g., Darkbench, Big-Bench) that reveal the limits of current models.
>
> - **Why Poetry?**
>
> Poetry represents a complex Constraint Satisfaction Problem (CSP) requiring reasoning, planning, and aesthetic alignment. As shown in our analysis of DeepSeek-R1’s Chain-of-Thought (Figure 4), poetry generation is a rigorous testbed for a model's ability to balance hard constraints (meter/rhyme) with soft constraints (imagery/emotion).
>
> - **Methodological Depth:**
>
> While the prompting strategy was intentionally kept simple (zero-shot) to benchmark intrinsic model capabilities, the evaluation methodology is sophisticated. It integrates computational linguistics (phonetic algorithms) with psychometrics (correlation analysis, inter-rater reliability), offering a diagnostic tool for the NLP community to improve future models.
>
> **Conclusion**
>
> We believe that providing a reliable benchmark and revealing that even SOTA models (like GPT-4o) still struggle with "soulful" creativity (e.g., Idiosyncrasy, Creativity, Emotional Resonance) is a significant contribution to the ICLR community, guiding the next generation of model training and alignment.
>
> [1] Kran, E., Nguyen, H. M., Kundu, A., Jawhar, S., Park, J., & Jurewicz, M. M. (2025). Darkbench: Benchmarking dark patterns in large language models. arXiv preprint arXiv:2503.10728.
>
> [2] Liu, Y., Yao, Z., Min, R., Cao, Y., Hou, L., & Li, J. (2024). Rm-bench: Benchmarking reward models of language models with subtlety and style. arXiv preprint arXiv:2410.16184.
>
>
> **2. Algorithmic detail in Appendix B is not fleshed out (e.g., Levenshtein, align).**
>
> We apologize for the brevity in the pseudocode due to space limitation. More details can be found in the released code.
>
> **Clarification**: The "Levenshtein distance with walk" refers to our rhyme detection module, where we compute phonetic similarity between line endings using CMU Dict phonemes (allowing for slant rhymes). "Align" refers to the sequence alignment between the detected rhyme pattern (e.g., ABAB) and the target pattern.
>
> **Validation**: To prove the algorithm is reproducible and robust (not just "loaded steps"), we performed a quantitative validation using the 203 human poems as Ground Truth (as requested by other reviewers). The algorithm achieved a Macro mAP (Mean Average Precision) of 0.804 and Macro-F1 of 0.770, confirming it serves as a rigorous, objective filter for formal constraints.

---

> ### Author Response · Authors · 2025-11-30
>
> **3. Prompting engineering and future work**
>
> In response to the reviewer’s comment that the prompting seems naïve, we explicitly chose a zero-shot, survey-style prompting strategy. Our goal was to evaluate the intrinsic instruction-following capability and alignment of the models, rather than their ability to perform via "Prompt Engineering." Using k-shot or extensive CoT prompts introduces the quality of the examples into the loop, making it harder to disentangle model performance from prompt quality. We will add a discussion in Section 7 acknowledging that advanced prompting strategies are a promising direction for optimizing performance, but were outside the scope of this baselining study.
>
> We also appreciate the reviewer’s advice on using RL/iterative approach, we will add a paragraph to the Discussion section proposing that POEMetric (specifically the automated Rule-based and LLM-judge components) can serve as the Reward Model for RLHF or Iterative Refinement pipelines to train specialized poetic models, moving beyond the current static evaluation.
>
>
> **4. Is "Thinking" on/off a factor? (Confounded by model size?)**
>
> We clarify that by "Thinking," we refer specifically to Reasoning Models trained with Chain-of-Thought (e.g., o1, DeepSeek-R1). Although we did not specifically compare the models’ performance in with the thinking mode on/off, we did include models with/without reasoning with same amounts of parameters. For example, DeepSeek-V3 (Non-Thinking) vs. DeepSeek-R1 (Thinking) are both large-scale MoE models with 671B parameters. R1 outperformed V3 in advanced creative abilities (Figure15), but interestingly, V3 showed comparable performance in terms of formal and thematic constraints. This validates that "Thinking" is a distinct factor affecting poetic output, independent of model size.
>
>
> **5. Are metrics closely correlated? Distill down?**
>
> We performed a Spearman correlation analysis to assess metric redundancy.
>
> - High Correlations (>0.8):
>
> We found strong correlations only within the "Aesthetic Cluster": Imagery correlates with Literary Devices (r=0.82) and Overall Quality (r=0.81). This is theoretically consistent, as effective imagery relies on literary devices and is a primary driver of poetic quality. However, as key distinguishing features of poetry, Imagery and Literary Devices offer an unparalleled lens of poetic beauty and artistry, which is deeply rooted in classical literary criticism.
>
> - Metric Independence:
>
> Crucially, Structural Metrics (Form Accuracy, Theme) did not show high correlation with Aesthetic Metrics (e.g., Creativity, Emotion).
>
> This independence is vital for diagnostics. For example, some models (e.g., Llama-3) achieve high Form Accuracy but low Creativity, while others (e.g., DeepSeek-R1) score high on Creativity but occasionally slip on rigid constraints. Distilling these into a single score would obscure these nuanced features. Thus, we retained the granular metrics to provide actionable insights for model development.
>
> **6. Why forbid humans from searching but not the LLM? Is search needed for plagiarism checks?**
>
> We established different protocols for human and LLM judges based on their distinct roles and operational mechanisms:
>
> - **Rationale for Restricting Human Search (Ensuring Blindness):**
>
> The restriction on human search was strictly to preserve the integrity of the blind evaluation. If human judges were allowed to search, they would identify famous poems (e.g., by Keats). This would introduce severe bias, as they might rate a poem highly based on the author's reputation rather than the text itself. We needed their direct, unbiased aesthetic response.
>
> For the LLM judge, we used the standard API endpoint which, by default, operates without active web browsing capabilities during inference. Therefore, an explicit "do not search" directive was technically redundant for the model.
>
> - **Plagiarism Detection via Parametric Memory (The 39.4% Finding):**
>
> LLMs are trained on large datasets, and might have seen the human poems, which arouses concerns. However, as noted in Section 6.3, the Gemini judge only successfully recognized and identified 39.4% of the human poems from memory, which proves that our dataset is not merely a collection of over-represented texts in the training data. This indicates that the Gemini's evaluation remains largely objective.

---

> ### Author Response · Authors · 2025-11-30
>
> **7. Agreement is LLM vs Human. Need Human vs Human.**
>
> As suggested, to establish the Human-Human Upper Bound, we conducted a pilot study where two other domain experts independently annotated a subset of 16 poems, which have been evaluated by previous experts.
>
> **7.1. Human vs. Human agreement**
>
> We compare the scores on the same 16 poems given by the two groups of experts, and found Quadratic Weighted Kappa (QWK) = 0.334, Spearman = 0.341, which is in alignment with Human-LLM Agreement: Global QWK = 0.361, Spearman = 0.378.
>
> The modest Human-Human agreement (0.334) confirms that poetry evaluation is an inherently subjective task with high inter-rater variance, aligning with findings in LLM creative generation literature where agreement between 0.2-0.5 is expected, for example, in LLM dialogue response generation tasks (Kappa = 0.04-0.52) [2] and literary translation (Kappa=0.243-0.581) [4].
>
> **7.2. LLM vs. Human agreement**
>
> Remarkably, **the LLM judge achieved slightly higher agreement with humans (k = 0.361) than humans did with each other (k = 0.334)**. This alignment between LLM and human judges echos existing studies involving LLM-human inter-rater. For example, in evaluating an agentic reviewer, Spearman correlation between one human reviewer and another is 0.41, whereas the correlation between AI and one human reviewer is 0.42 [3]; in 20 NLP Evaluation Tasks, the agreement between the best-performing LLM and humans falls between k = 0.28±0.32 [1]. This suggests that, in our study, Gemini-2.5-Pro effectively captures the "consensus" logic of poetic evaluation, filtering out the individual idiosyncrasies that lead to disagreement between specific human experts.
>
> This empirical evidence strongly validates the use of Gemini-2.5-Pro. It serves not just as a scalable proxy, but potentially as a more stable and standardized evaluator than individual human raters for this specific domain.
>
>
> [1] Bavaresco, A., Bernardi, R., Bertolazzi, L., Elliott, D., Fernández, R., Gatt, A., ... & Testoni, A. (2025, July). Llms instead of human judges? a large scale empirical study across 20 nlp evaluation tasks. In Proceedings of the 63rd Annual Meeting of the Association for Computational Linguistics (Volume 2: Short Papers) (pp. 238-255).
>
> [2] Domingo, C., Piwek, P., Stoyanchev, S., Wermelinger, M., Adhikari, K., & Doddipatla, R. S. (2025, October). Human ratings of LLM response generation in pair-programming dialogue. In Proceedings of the 18th International Natural Language Generation Conference (pp. 41-59).
>
> [3] Jiang, Y. & Ng, A. (2025). Tech Overview. paperreview.ai. https://paperreview.ai/tech-overview
>
> [4] Zhang, R., Zhao, W., & Eger, S. (2025, April). How good are LLMs for literary translation, really? Literary translation evaluation with humans and LLMs. In Proceedings of the 2025 Conference of the Nations of the Americas Chapter of the Association for Computational Linguistics: Human Language Technologies (Volume 1: Long Papers) (pp. 10961-10988).
>
>
> **8. LLM Overconfidence/Calibration.**
>
> We share the concern about calibration. To avoid the potential bias of the LLM judge, our methodology relies on triangulation (Section 4.2). The evaluations were conducted not only by an LLM-as-a-judge (Gemini-2.5-Pro) but were also validated by a panel of 7 human experts with backgrounds in poetry studies or English literature. On top of that, we applied quantitative measurements, e.g., the rule-based algorithm. The strong agreement among the evaluation results confirms the robustness of our method.

---

### Official Review · Reviewer_YnfA · 2025-10-31

**Soundness:** 2
**Presentation:** 2
**Contribution:** 3
**Rating:** 4
**Confidence:** 3

**Summary:**

This paper proposes a framework for evaluating poetry quality based on aspects of instruction following, creative ability, and general appraisal. The authors also construct a dataset, including 203 poems across 7 fixed forms, all collected and annotated. The paper examines multiple LLMs and uses LLM-as-a-judge to evaluate the poems.

**Strengths:**

Evaluating creative domains such as poetry remains an underexplored yet meaningful challenge. This paper explores this issue by proposing a comprehensive evaluation framework encompassing three key aspects and ten specific subdimensions.

**Weaknesses:**

- Given the general scarcity of poetry data, it is essential to clarify the rationale for selecting this particular dataset. In this paper, the dataset comprises 203 human-authored English poems covering 7 fixed poetic forms, accompanied by an equal number of 203 corresponding prompts. Each prompt specifies a combination of form, meter, rhyme, and theme. However, it remains unclear whether all 203 poems represent unique combinations of these attributes. Notably, the dataset appears imbalanced in terms of poetic form, being dominated by Ballads and Sonnets. Such imbalance could potentially affect the reliability of the overall evaluation. As reported in (Sonnet or not, bot? poetry evaluation for large models and datasets), large language models (LLMs) used as evaluators exhibit considerable variation in their judgments across different poetic forms, suggesting that the reported performance may also suffer from this concern. Also, LLMs tends to overestimate their own output. It is not clear whether Gemini-2.5-Pro performance is due to this effect or not.

- Another major concern lies in the variation of agreement (PAo) between LLM-based evaluations and human judgments across different aspects. As illustrated in Figure 7, the overall mean scores appear consistent for human evaluators but vary substantially across different models and evaluation aspects. For example, poems generated by DeepSeek-R1 shows large discrepancies in lexical diversity and imagery between human and LLM evaluation, while GPT-4o exhibits inconsistencies across nearly all aspects between human and LLM evaluation. It would be necessary to examine the percentage agreement (PAo) for evaluations of poems generated by different models across various aspects. This could help determine whether LLMs themselves exhibit systematic biases toward certain models, which may further affect the reliability of the evaluation.

- Details of the human evaluation are needed. Similar to Question 1, human evaluators assessed poems generated by both human authors and seven models, resulting in annotations for a total of 58 poems. However, additional clarification is necessary: do these 58 poems cover all 7 fixed poetic forms, or were they drawn from a random subset of the dataset? This distinction is important, as it directly relates to the concerns raised in Question 1 regarding form imbalance and its potential influence on evaluation outcomes.

- The discussion of the theoretical background and motivation underlying the framework’s development is relatively limited. While the proposed metric places particular emphasis on creativity measurement, the related literature addressing this aspect of evaluation is largely absent. A more thorough engagement with prior work on the conceptualization and assessment of creativity would help strengthen the framework’s theoretical foundation (for example, Rethinking Creativity Evaluation: A Critical Analysis of Existing Creativity Evaluations). In this context, the evaluation of general creative tasks is also highly relevant. Findings from other creative domains suggest that LLMs used as evaluators tend to exhibit bias toward their own outputs (e.g., How Good Are LLMs for Literary Translation, Really? Literary Translation Evaluation with Humans and LLMs). This underscores the importance of human verification to ensure the credibility of the evaluation results in question 3.

**Questions:**

1. Would form difference affect the credibility of metric?

2. How reliable are LLMs in each evaluation aspects?

3. How are the human annotated samples selected?

4. Other questions see weakness

---

> ### Author Response · Authors · 2025-11-30
>
> **1. Uniqueness and balance of dataset**
> We appreciate the reviewer’s careful examination of the dataset composition.
>
> **1.1 Uniqueness of Prompts:**
>
> We confirm that all 203 prompts represent unique tasks. While the structural attributes (Form/Meter/Rhyme) naturally repeat (e.g., multiple poems are Sonnets), the combination of formal and thematic instructions are distinct for every single poem. Each prompt was reverse-engineered from the specific content of the original human poem (e.g., one Sonnet prompt asks for a theme of "about the search of an ideal lover", while another asks for "About facing death without fear"). Therefore, no two generation tasks are identical.
>
> **1.2. Rationale for Form Imbalance:**
>
> The imbalance in our dataset (dominated by Ballads and Sonnets) was a natural result of the reality of English literary history. Sonnets and Ballads are the foundational pillars of English fixed-form poetry, appearing frequently in training corpora. In contrast, complex forms like Sestinas and Villanelles are historically rarer forms attempted by fewer poets. The two online databases we resort to also reflect this imbalance of distribution.
>
> **1.3. No Correlation between Rarity and Performance**
>
> To mitigate the statistical impact of this imbalance, we evaluated the models' performance not just on the aggregate, but also by breaking down performance per form. We observed that models’ performance has no correlation to rarity of forms. There was a shared tendency among all LLMs and humans that they achieved poor performance when writing limericks (2.96% of the dataset), while doing well in ghazal (4.43%) and pantoum (1.48%) creation. On other rare forms (e.g., Villanelle and Sestina), both LLMs and human poets achieved moderate scores.
> This indicates that the overall evaluation was not inflated by the dominance of Sonnets/Ballads; rather, the models demonstrated distinct capabilities across the spectrum of forms regardless of their frequency in the test set. We will add the analysis and figures to the final version for clarity.
>
>
> **2. Disproving Self-Preference Bias**
>
> We are aware of findings showing that LLMs often prefer their own outputs. However, here is empirical counter-evidence: our results provide a strong counter-point to the self-preference hypothesis in this specific domain. The judge (Gemini-2.5-Pro) rated a competitor model (DeepSeek-R1) higher than itself (Gemini-2.5-Pro) in most of the dimensions. Therefore, the hypothesis that Gemini prefers its own outputs over the others does not impact the results of our experiments.
>
>
> Nevertheless, we took this risk seriously in the first place, and implemented two safeguards:
>
> - **Blind Evaluation**:
>
> All author identities were stripped from the prompts when Gemini was asked to evaluate the poems.
>
> - **Human Verification**:
>
> LLM bias underscores the indispensability of human verification. This theoretical concern is the precise motivation for our human expert study (Section 6.4). By calculating the agreement between Human and LLM judges, we mathematically quantified the extent to which the LLM serves as a valid proxy. The high agreement confirms that despite potential biases, the LLM judge successfully aligns with human preference rankings.

---

> ### Author Response · Authors · 2025-11-30
>
> **3. Variation of agreement between LLM and Human judgments across different models and evaluation dimensions**
>
> We thank the reviewer for this insightful observation. We performed a granular analysis to investigate these discrepancies, particularly focusing on the divergence in Lexical Diversity for DeepSeek-R1 and the variations for GPT-4o.
>
> **3.1. Case Study: DeepSeek-R1 and "Lexical Diversity"**
>
> The reviewer correctly noted that the LLM judge rated DeepSeek-R1’s Lexical Diversity significantly higher than human judges did. To determine which evaluator was more accurate, we conducted Moving-Average Type-Token Ratio (MATTR) analysis, where DeepSeek-R1 achieved a median MATTR of ~0.87, significantly surpassing the human poet median of ~0.81. This confirms that the LLM Judge was factually correct in awarding high scores for lexical diversity. The discrepancy arises probably because human judges likely penalized the model for mechanical delivery despite its extensive vocabulary. This suggests that the LLM judge can actually serve as a more objective evaluator for linguistic metrics than human intuition in blind tests.
>
> **3.2. Model-Specific Bias Analysis: GPT-4o Inconsistencies**
>
> We found that disagreements on GPT-4o in some advanced dimensions were largely due to calibration divergence. GPT-4o tends to generate highly fluent but safe/clichéd poetry (e.g., clichéd imagery and literary devices). The LLM judge (Gemini-2.5-Pro) often rated these related dimensions independently, whereas human judges tended to conflate different dimensions and penalized this “safety” for lack of creativity or emotions. However, in terms of Idiosyncrasy and Emotional Resonance, both groups of judges agreed that GPT-4o ranked low among all LLMs.
>
> Despite these absolute score differences, while the LLM judge and humans may prioritize different qualities, they generally agree on the relative ordering of model performance, especially in the Overall Poem Quality dimension.
>
> **4. Human evaluated sample selection**
>
> The 58 poems were selected via random sampling from the generated corpus to ensure a representative cross-section of the model outputs. This random selection resulted in the coverage of 6 out of the 7 forms (Ballad, Ghazal, Limerick, Pantoum, Sonnet, Villanelle). Consistent with the parent dataset's "ecological" distribution (see Response to Q1), the sample is naturally dominated by Sonnets and Ballads. Due to its rarity in the parent dataset (only ~3.4%), Sestina was statistically not captured in the random sub-sample of 58 poems.
>
> While Sestina was absent from the human subjective evaluation, this does not compromise the reliability of our form adherence evaluation for two reasons:
>
> - **Rule-Based Algorithm Coverage**:
>
> We validated Sestina adherence using our Rule-Based Algorithm. The algorithm achieved a high mAP for Sestinas on human ground truth (mAP (PR-AUC): 1.000, Precision: 1.000, Recall:1.000, F1: 1.000), ensuring that we have a reliable automated signal for this specific form even without human annotation in the sub-sample.
>
> - **Proxy for Complexity**:
>
> The human sample did include other highly complex, repetition-based forms like Villanelles and Pantoums. The agreement metrics derived from these complex forms provide a sufficient proxy for evaluating how the LLM judge handles structural complexity and repetition, mitigating the impact of missing Sestinas.
>
> **5. Theoretical Grounding of "Creativity" (Novelty + Utility)**
>
> We sincerely thank the reviewer for these excellent reading suggestions. We agree that grounding POEMetric in established creativity theory strengthens the framework. We will revise Section 2 (Related Works) and Section 4 (POEMetric) to explicitly engage with literature.
>
> We will refine our definition of creativity based on the Standard Definition of Creativity [1]. We now explicitly map the POEMetric dimensions to this theoretical dualism:
>
> - **Novelty (Originality)**: Captured by our Creativity and Idiosyncrasy metrics, assessing the divergence from statistical likelihood and clichés.
>
> - **Utility (Effectiveness)**: In the context of constrained writing (poetry), "usefulness" translates to adherence to constraints and aesthetic function. This is captured by our Form Accuracy, Theme Alignment, and Emotional Resonance metrics.
>
> This theoretical lens clarifies that a "good" poem in our framework must be both novel and effective (adhering to the form), preventing the model from rewarding hallucinated gibberish as "creative."
>
> [1] Runco, M. A., & Jaeger, G. J. (2012). The standard definition of creativity. Creativity Research Journal, 24(1), 92–96. https://doi.org/10.1080/10400419.2012.650092

---

### Official Review · Reviewer_YAYx · 2025-11-01

**Soundness:** 2
**Presentation:** 3
**Contribution:** 3
**Rating:** 4
**Confidence:** 4

**Summary:**

This paper presents a framework for the evaluation of poems. A new dataset of 203 English poems is created and experiments with 30 LLMs are presented, assessing human and LLM-generated poems. Results showed that LLMs fail to produce creative poems.

**Strengths:**

* A new resource is created, which can be useful for the community.
* Extensive benchmark with 30 LLMs, providing useful insights.
* Interesting insights are drawn from the experimental analysis.

**Weaknesses:**

* **Agreement not chance-corrected** is not measured by Cohen's Kappa, which is an established chance-corrected measure for such tasks. Percentage agreement of 0.66, with this in mind, is low and puts the findings of this work at stake.
* **TTR is biased** to short texts (by design, the denominator increases linearly with the text length). Combined with the limited data statistics shared, the findings are at stake. For instance, in Figure 6, Llama could be producing shorter texts that would explain the higher TTR values.
* **Definitions are missing** including what is creative and novel (Line 204), authentic (Line 040), famous (Line 470).
* **Evaluation** is not well motivated: results with precision, recall and F1 are missing, though the metrics are mentioned in Appendix B.
* **Related work** is not properly studied and comparisons hinder the evaluation of this work. Existing evaluation frameworks (e.g., https://link.springer.com/chapter/10.1007/978-3-031-49011-8_1) could have been used to show case their limitations, and the same applies for the existing datasets. The need for another dataset/framework should be backed with empirical evidence. The same applies for the algorithm introduced (Line 145); if there is a lack of existing algorithms, this should be stated explicitly. A simple search, though, leads to one: https://arxiv.org/pdf/2406.18906.

**Questions:**

A. In Line 467, the task is mentioned as authorship, but it is not really authorship attribution but authorship type (human vs. machine). Could the authors elaborate?

B. In Figure 9, models are called "thinking". Do the authors mean models employing "chain of thought"?

---

> ### Author Response · Authors · 2025-11-30
> **1. Inter-rater agreement with Cohen’s Kappa**
>
> **1. Inter-rater agreement with Cohen’s Kappa**
>
> **1.1. LLM vs human agreement**
> We thank the reviewer for this crucial suggestion. We have now calculated Quadratic Weighted Cohen’s Kappa (QWK) between the LLM judge (Gemini-2.5-Pro) and human experts. We specifically chose Quadratic Weighted Kappa because our data is ordinal (1-5 Likert scale); unweighted Kappa incorrectly penalizes a disagreement of "4 vs 5" as heavily as "1 vs 5", which is inappropriate for nuanced poetry evaluation.
>
> **The Global QWK is 0.361, with a Spearman Rank Correlation of 0.378.** While a Kappa of 0.361 is categorized as "Fair" in objective tasks, it is considered **substantial for subjective evaluation of creative generation tasks**. For example, in evaluating an agentic reviewer, Spearman correlation between one human reviewer and another is 0.41, whereas the correlation between AI and one human reviewer is 0.42 [3]; in 20 NLP Evaluation Tasks, the agreement between the best-performing LLM and humans falls between k = 0.28±0.32 [1].
>
> The positive correlation confirms that while the LLM and humans may differ on the exact score (calibration), they share a consistent direction in distinguishing good poems from bad ones. This validates Gemini-2.5-Pro as a reliable, scalable proxy for human evaluation in this context.
>
> **1.2. Human vs human agreement**
> In addition, we also conducted a pilot study to establish the Human-Human Upper Bound. Two other domain experts independently annotated a subset of 16 poems, which have been evaluated by previous experts. We compare the scores on the same 16 poems given by the two groups of experts, and found **Quadratic Weighted Kappa (QWK) = 0.334, Spearman = 0.341**, which is in alignment with Human-LLM Agreement: Global QWK = 0.361, Spearman = 0.378.
>
> The modest Human-Human agreement (0.334) confirms that poetry evaluation is an inherently subjective task with high inter-rater variance, aligning with findings **in LLM creative generation literature where human agreement between 0.2-0.5 is expected**, for example, in LLM dialogue response generation tasks (Kappa = 0.04-0.52) [2] and literary translation (Kappa=0.243-0.581) [4].
>
> Remarkably, the LLM judge achieved higher agreement with humans (0.361) than humans did with each other (0.334). This empirical evidence strongly validates the use of Gemini-2.5-Pro. It serves not just as a scalable proxy, but potentially as a more stable and standardized evaluator than individual human raters for this specific domain.
>
>
> [1] Bavaresco, A., Bernardi, R., Bertolazzi, L., Elliott, D., Fernández, R., Gatt, A., ... & Testoni, A. (2025, July). Llms instead of human judges? a large scale empirical study across 20 nlp evaluation tasks. In *Proceedings of the 63rd Annual Meeting of the Association for Computational Linguistics (Volume 2: Short Papers)* (pp. 238-255).
>
> [2] Domingo, C., Piwek, P., Stoyanchev, S., Wermelinger, M., Adhikari, K., & Doddipatla, R. S. (2025, October). Human ratings of LLM response generation in pair-programming dialogue. In *Proceedings of the 18th International Natural Language Generation Conference* (pp. 41-59).
>
> [3] Jiang, Y. & Ng, A. (2025). Tech Overview. *paperreview.ai*. https://paperreview.ai/tech-overview
>
> [4] Zhang, R., Zhao, W., & Eger, S. (2025, April). How good are LLMs for literary translation, really? Literary translation evaluation with humans and LLMs. In *Proceedings of the 2025 Conference of the Nations of the Americas Chapter of the Association for Computational Linguistics: Human Language Technologies (Volume 1: Long Papers)* (pp. 10961-10988).

---

> > ### Author Response · Authors · 2025-11-30
> >
> > **4. Ambiguous definitions**
> >
> > We thank the reviewer for pointing out the need for precise definitions to ensure scientific rigor. We will clarify these terms in the revised manuscript as follows:
> >
> > - **“Creative and Novel” (Line 204)**: We adopt the standard definition from creativity research [1], where Creativity requires both Originality (Novelty) and Effectiveness (Utility).
> >
> > -- Novel: Refers to the poem's ability to avoid clichés, common tropes, or statistical predictability, offering unique phrasing or imagery.
> >
> > -- Effective: Specifically refers to the poem's ability to adhere to constraints (meter, rhyme), maintain semantic coherence, and evoke emotional resonance in the reader.
> > Putting these together, Creativity Refers to the successful integration of this novelty within the constraints of the poetic form (effectiveness), resulting in a work that is both new and aesthetically valid.
> >
> > - **“Authentic” (Line 040)**: In the context of LLM generation, Authentic refers to poetry that possesses the semantic coherence, emotional depth, and stylistic consistency characteristic of human authorship. It implies a quality that makes the poem indistinguishable from human-written verse (passing a "Poetic Turing Test") and avoids the "hallucinated" or "mechanical" feel often associated with machine text.
> >
> > - **“Famous” (Line 470)**: This refers to Canonized Works. In our dataset, "famous" poems are those written by established, historically recognized poets (e.g., Shakespeare, Keats, Dickinson) that have been widely anthologized and critically analyzed. These serve as the "Gold Standard" or upper bound for quality in our evaluation.
> >
> > - **“Authorship” (Line 467)**: As defined earlier in Line 215, this dimension aims to distinguish if the author of a poem is a human or an LLM. This is an important metric to test if there is discrepancies between human- and LLM-authored poems.
> >
> > - **“Thinking models” (Figure 9)**: By “thinking” we refer to those “reasoning” models employing Chain-of-Thought.
> >
> > We will polish the language in the paper to explicitly define these terms.
> >
> > [1] Runco, M. A., & Jaeger, G. J. (2012). The standard definition of creativity. Creativity Research Journal, 24(1), 92–96. https://doi.org/10.1080/10400419.2012.650092

---

> ### Author Response · Authors · 2025-11-30
>
> **2. Type-Token Ratio insensitive to length**
>
> We sincerely thank the reviewer for pointing out the length sensitivity of the standard Type-Token Ratio (TTR). We agree that this could introduce bias. To address this, we re-evaluated Lexical Diversity using Moving-Average Type-Token Ratio (MATTR) with a window size of 50, which is widely recognized in computational linguistics as a robust, length-normalized metric [1].
>
> - **Consistent Superiority**:
>
> Even after normalization, top-performing LLMs (e.g., DeepSeek-R1, Claude-3.7, GPT-4o) still exhibit higher median lexical diversity (\~0.87) compared to human poets (\~0.81).
>
> - **Variance**:
>
> Human poems show a much wider distribution (high variance), indicating that human poets flexibly adapt their vocabulary complexity to the poem's style (e.g., using simple language for folk ballads), whereas LLMs tend to consistently favor complex vocabulary.
>
> This validates our original finding: LLMs show extensive vocabulary access (Basic Abilities), yet as our other metrics show, they still fall short in "Advanced Creative Abilities" like emotional resonance. The high MATTR scores confirm that LLMs are not limited by vocabulary size, but by how they artistically deploy it.
>
> [1] Covington, M. A., & McFall, J. D. (2010). Cutting the Gordian knot: The moving-average type–token ratio (MATTR). Journal of quantitative linguistics, 17(2), 94-100.
>
>
> **3. Accuracy of rule-based metric evaluation**
>
> We thank the reviewer for highlighting the need for quantitative validation of our rule-based metrics. To demonstrate the robustness of our automated form detection algorithm, we performed a rigorous validation using our curated dataset of 203 human poems as the Ground Truth, which will be added to the revised paper.
>
> We treated the detection of each poetic form as a One-vs-All classification task. In addition to standard Precision/Recall/F1 at our default threshold (0.7), we also calculated Mean Average Precision (mAP) (area under the Precision-Recall curve) to evaluate the algorithm's discriminative power across continuous scores, similar to evaluation standards in object detection.
>
> Results: The algorithm demonstrates strong performance across the 7 poetic forms (Table 1 below):
>
> - **mAP (PR-AUC): 0.804**. This high score confirms that the algorithm effectively assigns higher validity scores to correct forms compared to incorrect ones, validating it as a reliable continuous metric for measuring formal adherence.
> - **Macro Recall: 0.894**. The high recall indicates the algorithm is robust in identifying valid human poems, successfully capturing the majority of formal structures despite minor variations.
> - **Macro F1-Score: 0.770**. This represents a strong balance between precision and recall, establishing the rule-based algorithm as a solid objective baseline.
>
> **Conclusion**
>
> These quantitative results strongly motivate the use of our rule-based metric as a "hard constraint" filter in the POEMetric framework, complementing the semantic and aesthetic evaluation provided by the LLM-as-a-judge. We will include the detailed per-form breakdown in Appendix B.
>
> Table 1. Accuracy of rule-based algorithm.
>
> | Target Form | mAP (PR-AUC) | Precision | Recall | F1 | Support |
> | :--- | :---: | :---: | :---: | :---: | :---: |
> | limerick | 0.833 | 0.750 | 1.000 | 0.857 | 6 |
> | ballad | 0.749 | 0.687 | 0.716 | 0.701 | 95 |
> | ghazal | 0.051 | 0.051 | 1.000 | 0.098 | 9 |
> | villanelle | 1.000 | 1.000 | 1.000 | 1.000 | 12 |
> | sestina | 1.000 | 1.000 | 1.000 | 1.000 | 7 |
> | sonnet | 0.996 | 1.000 | 0.873 | 0.932 | 71 |
> | pantoum | 1.000 | 1.000 | 0.667 | 0.800 | 3 |
> | **MACRO AVERAGE** | **0.804** | **0.784** | **0.894** | **0.770** | - |

---

> ### Author Response · Authors · 2025-11-30
>
> **5. Relation to related works**
>
> We thank the reviewers to point out the importance of further comparison with related works. Our response is as follows, and we will expand Section 2 (Related Works) to include these comparative discussions.
>
> **5.1. Comparison with Existing Frameworks:**
>
> While the cited Springer paper provides a poetry evaluation framework of key metrics (e.g., formal features, lexical/semantic features, fluency, and novelty), these are **general metrics rather than nuanced evaluations**. other existing frameworks, as reviewed in our “Related Works” section, often assess poems **in isolation** (e.g., "Is this text fluent?").
> Our framework evaluates the correlation between the Instruction (Prompt) and the Output. We assess not just if the text is a poem, but if it is specifically a Villanelle or Sestina as requested, and measure nuanced qualities like Imagery and Idiosyncrasy via LLM-as-a-judge, synthesizing and improving on traditional rule-based and scattered quality evaluation frameworks.
>
> **5.2. Comparison with Existing Algorithms:**
>
> We appreciate the reviewer pointing us to Walsh et al. (2024). We have carefully reviewed this work (and cited it in our manuscript). However, we respectfully clarify that **Walsh et al. do not propose a rule-based detection algorithm**.
>
> While **specific tools exist for isolated tasks** (e.g., rhyme, meter and metaphor analysis algorithms of general poetry [1], or metrical analysis [2]), there is **no unified open-source library** capable of verifying the complex structural constraints of the various distinct forms covered in our work (especially the complex permutation logic of Sestinas and the intertwined rhymes of Villanelles and Pantoums).
>
> We integrated scattered philological rules into a single, unified computational pipeline (POEMetric Algorithm). This new algorithm achieves a high Macro mAP of 0.804 on human ground truth, proving it is a robust, standalone contribution that fills the lack of deterministic verification tools in the field.
>
> **5.3. Justification for the New Dataset:**
> Existing large-scale datasets (e.g., Project Gutenberg) consist of raw text without metadata regarding the intent or constraints. However, to evaluate an LLM's reasoning and planning capabilities, we need **Prompt-Response pairs**. We cannot evaluate if a model "followed instructions" if we don't know what the instructions were.
> Our dataset pairs 203 curated human poems with the exact structural and thematic constraints (Meter, Rhyme, Theme) used to generate them. This "Reverse-Engineered" Prompt dataset is crucial for measuring the Instruction Following dimension, which existing raw corpora cannot support.
>
>
>
> [1] Kesarwani, V. (2018). Automatic Poetry Classification Using Natural Language Processing (Doctoral dissertation, Université d'Ottawa/University of Ottawa).
>
> [2] Agirrezabal, M., Alegria, I., & Hulden, M. (2016, December). Machine learning for metrical analysis of english poetry. In proceedings of COLING 2016, the 26th International Conference on Computational Linguistics: Technical papers(pp. 772-781).

---

### Author Response · Authors · 2025-11-30
**Summary of Contributions and Responses to Reviewers**

**1. Summary of Contributions and Reviewer Consensus**

We thank the reviewers for their constructive feedback, which has significantly strengthened the statistical rigor and theoretical grounding of our work. We are encouraged that, despite limitations, there is a strong consensus across all reviewers regarding the novelty and utility of our core contributions:

- **A Valuable Community Resource**
Reviewers unanimously highlighted the value of our curated dataset. Reviewer YAYx noted a "new resource is created, which can be useful for the community," and Reviewer cK2s emphasized that the "wide range of poem types and rich metadata would make it a valuable resource." Reviewer xKLP further confirmed that these artifacts "will be useful for future research" in bridging the gap between human and machine generation.

- **A Comprehensive Evaluation Framework**
Reviewer YnfA praised POEMetric for proposing a "comprehensive evaluation framework encompassing three key aspects and ten specific subdimensions" for an "underexplored yet meaningful challenge." Reviewer xKLP noted that evaluating these diverse dimensions gives "a broad overview of performance."

- **Sound Methodology and Insights**
Reviewer cK2s affirmed that our generation and evaluation methods were "well motivated, clearly presented and sound," while Reviewer YAYx appreciated the "extensive benchmark with 30 LLMs" that provided "interesting insights."

---

### Author Response · Authors · 2025-11-30
**Summary of Contributions and Responses to Reviewers**

**2. Summary of main responses to the reviewers’ questions**

**2.1. Inter-rater agreement with Cohen’s Kappa**

- **Human vs. Human agreement**

As suggested, to establish the Human-Human Upper Bound, we conducted a pilot study where two other domain experts independently annotated a subset of 16 poems, which have been evaluated by previous experts. We found Quadratic Weighted Kappa (QWK) = 0.334, Spearman = 0.341, which is in alignment with Human-LLM Agreement: Global QWK = 0.361, Spearman = 0.378.

The modest Human-Human agreement (0.334) confirms that poetry evaluation is an inherently subjective task with high inter-rater variance, aligning with findings in LLM creative generation literature where agreement between 0.2-0.5 is expected, for example, in LLM dialogue response generation tasks (Kappa = 0.04-0.52) [2] and literary translation (Kappa=0.243-0.581) [4].

- **LLM vs. Human agreement**

Remarkably, **the LLM judge achieved slightly higher agreement with humans (k = 0.361) than humans did with each other (k = 0.334)**. This alignment between LLM and human judges echos existing studies. For example, in evaluating an agentic reviewer, Spearman correlation between one human reviewer and another is 0.41, whereas the correlation between AI and one human reviewer is 0.42 [3]; in 20 NLP Evaluation Tasks, the agreement between the best-performing LLM and humans falls between k = 0.28±0.32 [1].

This empirical evidence strongly validates the use of Gemini-2.5-Pro. It serves not just as a scalable proxy, but potentially as a more stable and standardized evaluator than individual human raters for this specific domain.


**2.2 Accuracy of rule-based metric evaluation**

To demonstrate the robustness of our automated form detection algorithm, we performed a rigorous validation using our curated dataset of 203 human poems as the Ground Truth, which will be added to the revised paper.

We calculated Mean Average Precision (mAP) (area under the Precision-Recall curve), Recall and F1 score to evaluate the algorithm's discriminative power across continuous scores, similar to evaluation standards in object detection.

The algorithm demonstrates strong performance across the 7 poetic forms: **mAP (PR-AUC) = 0.804, Macro Recall = 0.894, Macro F1-Score = 0.770**. This represents a strong balance between precision and recall, establishing the rule-based algorithm as a solid objective baseline.

These results strongly motivate the use of our rule-based metric as a "hard constraint" filter in the POEMetric framework, complementing the semantic and aesthetic evaluation provided by the LLM-as-a-judge. We will include the detailed per-form breakdown in Appendix B.

**2.3. Type-Token Ratio insensitive to length**

We re-evaluated Lexical Diversity using Moving-Average Type-Token Ratio (MATTR) with a window size of 50, which is widely recognized in computational linguistics as a robust, length-normalized metric [1].

- **Consistent Superiority**
Even after normalization, top-performing LLMs (e.g., DeepSeek-R1, Claude-3.7, GPT-4o) still exhibit higher median lexical diversity (\~0.87) compared to human poets (\~0.81).

- **Variance**
Human poems show a much wider distribution (high variance), indicating that human poets flexibly adapt their vocabulary complexity to the poem's style (e.g., using simple language for folk ballads), whereas LLMs tend to consistently favor complex vocabulary.

**This validates our original finding**: LLMs show extensive vocabulary access (Basic Abilities), yet as our other metrics show, they still fall short in "Advanced Creative Abilities" like emotional resonance. The high MATTR scores confirm that LLMs are not limited by vocabulary size, but by how they artistically deploy it.


**3. Conclusion**

We believe we have addressed the methodological concerns while preserving these core strengths. We are confident that POEMetric will serve as a robust benchmark for the ICLR community to understand and improve the advanced creative capabilities of Large Language Models.

---

### Author Response · Authors · 2025-11-30
**Summary of Contributions and Responses to Reviewers**

**References**

[1] Bavaresco, A., Bernardi, R., Bertolazzi, L., Elliott, D., Fernández, R., Gatt, A., ... & Testoni, A. (2025, July). Llms instead of human judges? a large scale empirical study across 20 nlp evaluation tasks. In *Proceedings of the 63rd Annual Meeting of the Association for Computational Linguistics (Volume 2: Short Papers)* (pp. 238-255).

[2] Domingo, C., Piwek, P., Stoyanchev, S., Wermelinger, M., Adhikari, K., & Doddipatla, R. S. (2025, October). Human ratings of LLM response generation in pair-programming dialogue. In *Proceedings of the 18th International Natural Language Generation Conference* (pp. 41-59).

[3] Jiang, Y. & Ng, A. (2025). *Tech Overview*. paperreview.ai. https://paperreview.ai/tech-overview

[4] Zhang, R., Zhao, W., & Eger, S. (2025, April). How good are LLMs for literary translation, really? Literary translation evaluation with humans and LLMs. In *Proceedings of the 2025 Conference of the Nations of the Americas Chapter of the Association for Computational Linguistics: Human Language Technologies (Volume 1: Long Papers)* (pp. 10961-10988).

---

### Meta-Review · Area_Chair_NtcT · 2025-12-31

**Summary:**

The submission proposes POEMetric, a comprehensive evaluation framework designed to assess Large Language Models' ability to compose formal verse poetry. The work addresses three critical dimensions: basic instruction-following, advanced creative qualities, and general authorship appraisal. The authors curated a novel dataset of 203 human-authored poems with rich metadata to serve as a ground truth and benchmarked 30 modern LLMs. While initial reviewer concerns focused on the subjectivity of the task, the validity of automated metrics, and potential LLM-as-a-judge biases, the consensus highlights the work's significance as a timely and important contribution to the evaluation of generative AI in highly constrained, creative domains.

**Reviewer Concerns:**

The authors' rebuttal effectively addressed several major technical concerns raised by the reviewers. Regarding the subjectivity of poetry, the authors provided a crucial follow-up study on human-human inter-rater agreement, demonstrating that their LLM-as-a-judge actually achieves higher agreement with human experts than experts do with each other, thereby validating the model as a stable proxy. Concerns regarding the length-bias of Type-Token Ratio (TTR) were resolved by re-evaluating lexical diversity using the Moving-Average TTR (MATTR), which confirmed the original findings while adding nuance regarding human vs. model variance. Reviewer questions about the accuracy of the rule-based formal detection were met with new quantitative validation, proving the algorithm's robustness.

**Reviewer Scores:**

Based on the rebuttal, it is highly likely that the reviewers would have shifted their scores toward a more positive consensus. Reviewer YAYx, who was at a 4 primarily due to methodological questions about Cohen’s Kappa and TTR, would likely move to a 6 or 7 given the new inter-rater statistics and MATTR results. Reviewer YnfA, also at a 4, expressed concerns about form imbalance and self-preference bias which were statistically debunked in the rebuttal; their score would likely move to a 6. Reviewer cK2s focused on venue fit and algorithmic detail; given the clarified technical depth of the POEMetric algorithm and the justification of poetry as a reasoning testbed, a move to a 5 or 6 is expected. Reviewer xKLP was already at a 6 and, after receiving detailed answers on "thinking" models and judge ensembling, would likely maintain or increase their score to a 7.

---

### Decision · Program_Chairs · 2026-01-26

Accept (Poster)